
# Walking, Weak first-order transitions, and Complex CFTs II. Two-dimensional Potts model at $Q > 4$

**Victor Gorbenko[1], Slava Rychkov[2,3] and Bernardo Zan[2,3,4]**

**1** Stanford Institute for Theoretical Physics, Stanford University, Stanford, CA 94305, USA
**2** Institut des Hautes Études Scientifiques, Bures-sur-Yvette, France
**3** Laboratoire de physique théorique, Département de physique de l'ENS
École normale supérieure, PSL University, Sorbonne Universités,
UPMC Univ. Paris 06, CNRS, 75005 Paris, France
**4** Institut de Théorie des Phénomènes Physiques, EPFL, CH-1015 Lausanne, Switzerland

## Abstract

We study complex CFTs describing fixed points of the two-dimensional $Q$-state Potts model with $Q > 4$. Their existence is closely related to the weak first-order phase transition and the "walking" renormalization group (RG) behavior present in the real Potts model at $Q > 4$. The Potts model, apart from its own significance, serves as an ideal playground for testing this very general relation. Cluster formulation provides nonperturbative definition for a continuous range of parameter $Q$, while Coulomb gas description and connection to minimal models provide some conformal data of the complex CFTs. We use one and two-loop conformal perturbation theory around complex CFTs to compute various properties of the real walking RG flow. These properties, such as drifting scaling dimensions, appear to be common features of the QFTs with walking RG flows, and can serve as a smoking gun for detecting walking in Monte Carlo simulations.

The complex CFTs discussed in this work are perfectly well defined, and can in principle be seen in Monte Carlo simulations with complexified coupling constants. In particular, we predict a pair of $S_5$-symmetric complex CFTs with central charges $c \approx 1.138 \pm 0.021i$ describing the fixed points of a 5-state dilute Potts model with complexified temperature and vacancy fugacity.

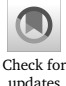
## 1 Introduction

The term 'walking' refers to RG flows described by the beta-function

$$\beta(\lambda) = -y - \lambda^2, \tag{1.1}$$

with $y > 0$ a small fixed parameter. The flow happens from $\lambda \sim -1$ to $\lambda \sim +1$ and slows down a lot when $\lambda \sim 0$, giving rise to an exponentially large hierarchy between the UV and IR scales, of the order $\exp(\pi/\sqrt{y})$.

In a recent paper [1] we reviewed this mechanism of generating hierarchies, and pointed out several examples of physical systems which realize it. One example are 4d and 3d gauge theories, where the walking mechanism is realized, conjecturally, below the lower end of the conformal window. Another example is the $Q$-state Potts model which has a conformal phase at $Q \leqslant Q_c(d)$, and the walking mechanism governs the physics of a weakly first-order transition just above $Q_c$. Abundant evidence, especially in $d = 2$ where $Q_c = 4$, allows to firmly establish walking in the Potts model.

Another goal of [1] was to highlight the concept of 'complex CFTs', an unusual class of conformal field theories which describe fixed points of RG flow (1.1) occurring at complex coupling $\lambda = \pm i\sqrt{y}$. These complex CFTs control walking RG flow passing near them, in a way similar to how a UV fixed-point CFT controls the beginning of the RG trajectory arising from it via a relevant perturbation.

This second paper of the series will develop further the connection between walking and complex CFTs, by studying in depth the 2d Potts model at $Q > 4$. While CFTs describing the conformal phase of the 2d Potts model at $Q \leqslant 4$ have been studied intensely [2–4], as far as we know, our work is the first one discussing the complex CFTs at $Q > 4$.

We start in section 2 reviewing 2d Potts model results relevant for our purposes. We discuss the spin and cluster formulations of the model, transition to the height representation, and the Coulomb gas construction for the conformal phase at $Q \leqslant 4$. We present the partition

function of the model on the torus, and obtain from it the spectrum of low-lying operators. We also speculate about possible implementations of the permutation symmetry with non-integer number of elements.

In section 3 we analytically continue the conformal Potts theory in $Q$ to real $Q > 4$, which leads to complex conformal theories. In section 4 we reconstruct the real $Q > 4$ theory at the first-order phase transition by means of conformal perturbation theory around the complex CFT. We present several one- and two-loop consistency checks, and in particular compute drifting scaling dimensions – a characteristic feature of two-point functions in theories exhibiting walking RG behavior.

Several technical points are relegated to the appendices. Appendix A presents our version of the argument which relates the Coulomb gas coupling constant to $Q$. Appendix B reviews some representation theory of the permutation group and discusses a few operators transforming in its higher representations. Appendix C reviews some basic results in the orbifold construction of the $Q = 4$ theory.

## 2  2d Potts model for $Q \leqslant 4$

In this paper we deal with the 2d Potts model with $Q$ states. An elementary introduction to this model was already given in [1]. Here we will repeat definitions for completeness and provide a few further details. Almost everything we say in this section will be well known to the experts on the 2d Potts model and the related loop models. Still, by our own experience it's not always easy to parse the results scattered throughout the literature, so we will provide a self-contained exposition of the needed facts. One place where our point of view differs from the existing literature is concerning Kondev's argument, see footnote 16 and appendix A.

Consider first the lattice formulation, working on a square lattice for definiteness. For integer $Q$ we have a model of spins $s_i$ living on the lattice sites, which can be in $Q$ states labeled $1, \ldots, Q$, and which have ferromagnetic nearest-neighbor interaction $-\beta \delta_{s_i, s_j}$, preserving $S_Q$ global symmetry.

This model can be rewritten in terms of probability distribution of random graphs $X$ living on the same lattice (called the Fortuin-Kasteleyn, or cluster representation). The graphs $X$ include all lattice sites and some of the bonds, and the weight of a given graph is given by $v^{b(X)} Q^{c(X)}$ where $b(X)$ is the number of bonds and $c(X)$ is the number of connected components (clusters). The two definitions give an identical partition function for integer $Q$ if $v = e^{\beta} - 1$. Notably, the second definition also makes sense for non-integer $Q$ and allows to analytically continue the model in $Q$. Here we will consider real $Q > 0$.

The model has an order-disorder phase transition located at $v = \sqrt{Q}$, which is continuous for $Q \leqslant Q_c = 4$. The CFT describing this transition is called the critical Potts model.

One can also define the dilute Potts model where certain sites of the lattice are kept vacant. Varying both the fugacity of the vacancies and the temperature, the dilute model has a tricritical point for $Q \leqslant 4$. (It also has a critical point which is the same as for the original Potts model.) The CFTs describing the tricritical and critical Potts model are different for $Q < 4$ and coincide for $Q = 4$.

For integer $Q = 2, 3, 4$ the CFTs describing the tricritical and critical Potts model are unitary and exactly solvable (for $Q = 2, 3$ these are unitary minimal models, while for $Q = 4$ it's an orbifold of compactified free boson, see appendix C). For non-integer $0 < Q < 4$ as well as

$Q = 1$ these CFTs are real (in the sense that all the observables are real, see section 6 of [1]) but non-unitary. As we will see below, the spectrum of local operators of the tricritical and critical Potts models is exactly known for all $0 < Q \leqslant 4$. However, not all the OPE coefficients among these operators are exactly known for $Q$ different from 2,3,4. So these CFTs have not been exactly solved.[1]

## 2.1 Lattice transfer matrix and local operators

The cluster definition of the Potts model for non-integer $Q$ being nonlocal, it may seem puzzling how a local CFT may describe its phase transition. This section will clarify the physical meaning of CFT operators in the cluster definition. This discussion is a bit technical, and the reader interested merely in the applications of the Potts model to the subject of complex CFTs, rather than in the physics of the Potts model itself, can skip to Eq. (2.18) where we begin to present the results for the torus partition function.

Recall that in the familiar integer $Q$ case, when we can describe the Potts model in terms of spins, local lattice operators are obtained by fixing the values of a certain number of spin variables at nearby points. A general lattice operator will have the form

$$\delta(s(x_1) = a_1) \cdots \delta(s(x_n) = a_n), \tag{2.1}$$

where $s(x_1), \ldots, s(x_n)$ are spin variables at nearby lattice points and $a_1, \ldots, a_n$ are fixed values. These operators can be further grouped into irreducible representations (irreps) of $S_Q$ symmetry (see e.g. [7]).

In this case, the correspondence between lattice operators and local CFT operators is as follows. At the critical point we have local CFT operators with well-defined scaling dimension and spin, transforming in an irrep of $S_Q$. Each lattice operator can be expanded into CFT operators. Going in the opposite direction, for each local CFT operator we can find a local lattice operator so that their correlation functions agree at distances large compared to the lattice spacing.

Another way to make contact between the lattice and the CFT is to consider the model on the cylinder $S^1 \times \mathbb{R}$, i.e. with one dimension compactified on a circle of length $L$, and the other thought of as (Euclidean) time. States propagating along the cylinder have energies $(2\pi/L)\Delta_i$, where $\Delta_i$ are scaling dimensions of the CFT operators (up to a constant shift $-c/12$ due to the conformal anomaly). These energies can be measured on the lattice by constructing the transfer matrix and measuring its eigenvalues (see below).

Going next to the cluster description applicable also for non-integer $Q$, the simplest nontrivial observable is the probability that two distant points $x$ and $y$ are in the same cluster. This probability is the cluster analogue of the spin-spin correlation function. One can also consider more complicated events, e.g. probability that two groups of $n$ nearby points $x_1, \ldots, x_n$ and $y_1, \ldots, y_n$ belong pairwise to $n$ different clusters. Such probabilities are cluster analogues of two-point (2pt) functions of operators made of several spins for integer $Q$ (see e.g. [8]). So, roughly, a local operator creates a localized disturbance in the cluster distribution. One type of disturbance is to emit a certain number of clusters. For integer $Q$, these operators correspond in the spin description to operators transforming nontrivially under $S_Q$. On the other hand, disturbances which don't emit clusters correspond to operators which are singlets under $S_Q$.

---

[1]See [5] and [6] for recent progress on the $Q = 1$ case (percolation) using a numerical conformal bootstrap approach.

In local spin models, in particular in the Potts model for integer $Q$, it is standard to extract scaling dimensions of local operators by analyzing eigenvalues of the transfer matrix $T_{\text{spin}}$ on a cylinder (i.e. on a lattice with periodic boundary conditions in one direction). Analogously, dimensions of scaling operators for non-integer $Q$ can be extracted from a transfer matrix $T_{\text{cluster}}$ in the cluster representation.[2] This $T_{\text{cluster}}$ differs from $T_{\text{spin}}$ in a few aspects, in particular they act on rather different spaces of states. The familiar $T_{\text{spin}}$ acts in the space of spin states in a given time slice $\tau$. On the other hand, $T_{\text{cluster}}$ acts in a vector space spanned by *connectivity states*, which refer to two time slices, the initial 0 and the final $\tau$, and are defined as follows. Suppose we already built the partition function on the cylinder from time 0 up to $\tau$ and we want to add another layer to the lattice $\tau \to \tau + 1$. To do this we only need to know how the $2L$ lattice sites $\{1, 2, \ldots, L\}$ at time 0 and $\{1', 2', \ldots, L'\}$ at time $\tau$ are connected among each other by clusters. A connectivity state is a partition $P$ of these $2L$ sites into groups connected by clusters. For example the situation in Fig. 1 corresponds to $P = \{\{2\}, \{1, 1', 3\}, \{2', 3'\}\}$. The transfer matrix $T_{\text{cluster}}$ maps state $P_\tau$ to a linear combination of states $P_{\tau+1}$, and can be constructed using two basic operations: join and detach. The join operation $J_{xy}$ joins two clusters to which $x, y$ belong, while the detach operation $D_x$ detaches point $x$ from its cluster (this process has weight $Q$ if $x$ was already by itself).

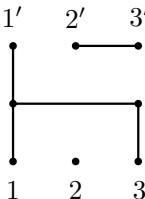

Figure 1: This graph corresponds to the connectivity state described by a partition $P = \{\{2\}, \{1, 1', 3\}, \{2', 3'\}\}$.

An important characteristic of a state $P$ is the number of bridges $b$, defined as clusters which connect time 0 and time $\tau$ ($b = 1$ in Fig. 1). Clearly, the number of bridges can either remain constant or decrease under the action of $T_{\text{cluster}}$, giving this transfer matrix an upper-block-triangular structure [11]. We are interested in the eigenvalues of $T_{\text{cluster}}$, which correspond, in the large $L$ limit, to the dimensions of local operators. To each eigenvalue $\lambda$ we can associate the maximal number of bridges present in the eigenvector, $b_\lambda$. To compute the eigenvalues, we can first diagonalize the blocks $T_b$ of the transfer matrix which leave the number of bridges constant and equal to $b$. The full transfer matrix will then have the same eigenvalues, the part of the eigenvectors with $b' < b$ uniquely reconstructable from the part with $b$ bridges using the block-triangular structure.[3]

In this language, the eigenvalues with $b_\lambda = 0$ correspond to the 'singlet' sector of the theory. For integer $Q$, these eigenvalues correspond to states which are singlets under $S_Q$. From the leading eigenvalue we extract the ground state energy (the central charge), from the subleading ones the dimensions of operators $\varepsilon$ and $\varepsilon'$ which will appear below, etc. On the other hand, the eigenvalues with $b_\lambda > 0$ correspond to operators 'transforming nontrivially under the symmetry' (borrowing classification from integer $Q$). For example, the spin operator

---

[2]This formalism goes back to [9]. Our discussion is based on [10–12]. We are grateful to Jesper Jacobsen for explanations.

[3]Here we are assuming that eigenvalues are non-degenerate among different blocks $T_b$. Otherwise the full transfer matrix may not be diagonalizable, rather reducible to a Jordan normal form. This more complicated situation corresponds in the continuum limit to logarithmic CFTs. It is realized in the limit $Q \to 1$ describing percolation [13].

will correspond to the first eigenvalue in the sector with $b_\lambda = 1$.

Let us discuss briefly how the transfer matrix is used to compute the Potts partition function on a torus, i.e. with periodic boundary conditions in both directions. At integer $Q$, the partition function is given by

$$Z = \text{Tr}(T_{\text{spin}})^N, \tag{2.2}$$

where $N$ is the time-direction extent of the torus. In terms of transfer matrix eigenvalues $\lambda$ this can be written as

$$Z = \sum_\lambda M_\lambda \lambda^N, \tag{2.3}$$

where $M_\lambda$ are integer eigenvalue multiplicities. At integer $Q$ we have $S_Q$ symmetry, and $M_\lambda$'s are dimensions of (possibly reducible) representations of $S_Q$.

The analogue of (2.2) for non-integer $Q$ looks more complicated:

$$Z = \text{Tr}\big[G(T_{\text{cluster}})^N\big], \tag{2.4}$$

where $G$ is a gluing operator which makes the torus out of the cylinder and takes into account that clusters can be reconnected nontrivially during this operation. Thus we can still write (2.3), but now $M_\lambda$'s are products of eigenvalue multiplicities times matrix elements of the gluing operator. In particular, $M_\lambda$ will not in general be an integer nor even positive. The coefficients $M_\lambda$ can be evaluated combinatorially [14] and are polynomials in $Q$. We will have more to say about them below when we will discuss the partition function in the continuum limit using the Coulomb gas method.

Notice that by construction Eq. (2.4) should agree with Eq. (2.2) for integer $Q$, although this is not manifest because the cluster transfer matrix is not obviously related to the spin transfer matrix. In fact for large integer $Q$, Eq. (2.4) provides a more efficient way to evaluate the partition function of the Potts model than Eq. (2.2), because the Hilbert space dimension is much smaller.

Finally, we note that while the above discussion focused on the Potts model, it can be adapted to the diluted Potts model by allowing for vacancies. In particular, it is possible to study operator dimensions of the tricritical Potts model by means of a cluster transfer matrix [15].

## 2.2 Symmetry

What is the symmetry of the Potts model? For integer $Q$, it's $S_Q$, while for non-integer $Q$ it should be some sort of analytic continuation of $S_Q$. As mentioned in [1], section 3.3, precise mathematical meaning of this symmetry can likely be given using the language of Deligne's tensor category $\text{Rep}(S_t)$ [16]. Transfer matrix of the Potts model will be a morphism in this category. Here as in [1] we will take an intuitive approach to symmetry for non-integer $Q$ – as something which exists and which will be clarified in future work. For example, we will try to expand partition function multiplicities into dimensions of representations of $S_Q$ analytically continued to non-integer $Q$, although clearly there is no such thing as a vector space of non-integer dimension. Another consequence of the symmetry is that $Q$ doesn't renormalize, even if non-integer. So $Q$ is viewed as a fixed parameter characterizing the theory, not as a coupling constant. This will be important when we study RG flows in perturbed Potts models. Readers bewildered by non-integer $Q$ may adopt the point of view that only integer $Q$ is 'physical', while the intermediate $Q$ is just a trick to do the analytic continuation. We don't endorse such a restricted point of view, but it can be a helpful crutch.

### 2.3 Height representation

The cluster representation was applicable to the Potts model in any number of dimensions. Here we will describe the loop and the height representations, which are specific for 2d. These representations are the key to the torus partition function calculation explained in the next section.

The loop representation is obtained by drawing loops surrounding clusters on the 'medial lattice' whose sites are midpoints of the bonds on the original lattice. Each loop is given weight $\sqrt{Q}$, and at the critical temperature the partition functions of the Potts model and of the loop model coincide (see e.g. [3], Eq. (4.5)), if one works on a finite lattice with free boundary conditions as in Fig. 2.

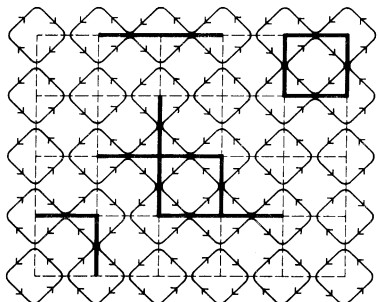

Figure 2: Passing from a cluster configuration to an oriented loop configuration (figure from [3]).

One then further passes from the loop model to a height model (also called solid-on-solid model). This is accomplished by splitting the loop weight into two terms corresponding to two possible loop orientations. Each loop orientation gets a complex weight $e^{\pm 4iu}$ obtained by multiplying factors $e^{\pm iu}$ for each left/right term (the total number of turns counted with sign being 4), and $u$ is chosen so that

$$\sqrt{Q} = 2\cos(4u), \tag{2.5}$$

summing over the two orientations. One then defines the height function $\phi$ starting from zero boundary condition and changing it by $\pm\phi_0$, where $\phi_0 = \pi/2$ by convention, when crossing any loop, so that larger height is always on the left of the arrow. The resulting height model, for $Q \leqslant 4$, is known to renormalize at long distances to the gaussian fixed point

$$\frac{g}{4\pi} \int d^2x \, (\nabla\phi)^2 \,, \tag{2.6}$$

where the coupling $g$ is related to $Q$ by

$$Q = 2 + 2\cos\frac{\pi g}{2}, \tag{2.7}$$

the branch $2 < g \leqslant 4$ chosen for the considered critical Potts model. This nontrivial result (see [2] for a review, as well as [3], Eq. (2.19)) is the foundation on which the rest of the construction is built.

For the tricritical Potts model the height representation is harder to build and we will not discuss it [17]. Once the dust settles, it turns out that the tricritical Potts model also renormalizes to the gaussian fixed point, the only difference being that one has to choose another solution branch of (2.7), namely $4 \leqslant g < 6$.

In summary, we have

$$g(Q) = 4 + \frac{2}{\pi} \cos^{-1}\left(\frac{Q-2}{2}\right), \tag{2.8}$$

with $\cos^{-1}$ in the interval $[-\pi, 0]$ on the critical and $[0, \pi]$ on the tricritical branches at $Q \leqslant 4$, see Fig. 3.

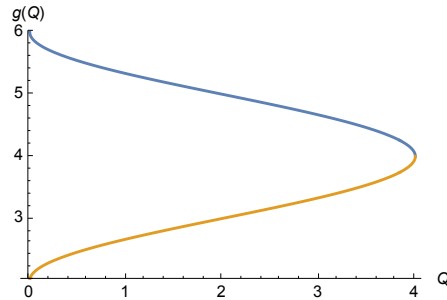

Figure 3: Coupling constant as a function of $Q \leqslant 4$ for the critical (lower branch, orange) and tricritical (upper branch, blue) Potts model.

$$\chi\quad\chi\quad\chi\quad\chi\quad\chi\quad\chi$$
$$w_1\quad w_2\quad w_3\quad w_4\quad w_5\quad w_6$$

Figure 4: Vertices of the 6-vertex model.

It might be surprising that the oriented loop model with complex weights led to the gaussian model (2.6) with real weights. One explanation is that we can map the oriented loop model on the $F$-model, which is a 6-vertex model with positive weights. The mapping consists in putting vertices on the medial lattice, with two arrows pointing inwards and two pointing outwards. The $F$-model vertices are shown in Fig. 4, with weights

$$w_1 = w_2 = w_3 = w_4 = 1, \qquad w_5 = w_6 = e^{2iu} + e^{-2iu}. \tag{2.9}$$

These assignments are in accord with the fact that vertices 1 to 4 can be uniquely decomposed into two loop strands (and $1 = e^{iu}e^{-iu}$), while for vertices 5 and 6 there are two possible decompositions, see Fig. 5. The height function for the $F$-model is the same as for the oriented loop model, and in the continuum limit the $F$-model renormalizes onto the free scalar boson (2.6).

$$)\,(\quad \Rightarrow \quad \chi$$
$$)\,(\; + \; \smile \quad \Rightarrow \quad \chi$$

Figure 5: Passing from the oriented loop model to the $F$-model.

In the above discussion we were a bit cavalier about the boundaries. Suppose we are working on a simply connected domain like a rectangle. Along the boundary, we have to use boundary vertices shown in Fig. 6, and decide which weight to assign to them. In the $F$-model, it is natural to give weight 1 to the boundary vertices, which is called free boundary conditions for the 6-vertex model, and corresponds to using the Dirichlet boundary conditions

for the corresponding height field. With this assignment, the $F$-model renormalizes onto the free scalar boson (2.6) with the Dirichlet boundary conditions.

On the other hand, to reproduce correctly the weight of the oriented loops, boundary vertices in the Potts model should be given weights $e^{\pm iu}$.[4] The product of these weights equals

$$e^{iu(C_+ - C_-)}, \tag{2.10}$$

where $C_\pm$ are parts of the perimeter occupied by left/right going loops. To illustrate the importance of this factor, consider the partition function. The $F$-model partition function will reduce in the continuum limit to the partition function of the free scalar boson (2.6) with Dirichlet boundary conditions. The Potts model partition function will be much more nontrivial, since we have to include factor (2.10) into the path integral.

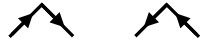

Figure 6: Boundary vertices, assigned weight 1 in the $F$-model, and weights $e^{\pm iu}$ in the Potts model.

## 2.4 Long cylinder partition function and the Coulomb gas

In the next section we will consider the torus partition function, where we won't have to deal with the above complications due to the boundary terms, but there will be other complications due to loops going around the torus. Here we would like to discuss partition function on the cylinder of circumference $L$ and length $T$. Were we to keep the rule that each oriented loop gets a weight obtained by multiplying $e^{\pm iu}$ for every left/right turn, we would get weight 1 instead of $e^{\pm 4iu}$ for loops circling around the cylinder, so that the unoriented loops get weight 2 instead of $\sqrt{Q}$. This discrepancy should be corrected as follows. Let us impose for definiteness the zero boundary condition on $\phi$ at the time 0 boundary of the cylinder. Then, on the lattice, at time $T$, $\phi$ will take a constant value given by $n(\pi/2)$ where $n$ is the number of oriented loops circling around the cylinder (counted with opposite sign for two opposite orientations). Changing the weight of such loops from 1 to $e^{\pm 4iu}$ can thus be accomplished multiplying the partition function with an extra factor

$$e^{ie_0\phi(T)}, \tag{2.11}$$

where $e_0 = 4u/(\pi/2)$, which gives

$$e_0 = 2 - g/2. \tag{2.12}$$

This is usually described as 'placing charges $\pm e_0$ at two opposite ends of the cylinder'. The whole construction is known as 'Coulomb gas'.[5]

To evaluate the Potts model partition function on the cylinder, we thus have to combine three ingredients: the $F$-model in the bulk which renormalizes to the free scalar boson, and the factors (2.10) and (2.11). For a finite aspect ratio $T/L$ this is a nontrivial task which

---

[4]We thank Hubert Saleur for explaining this point to us.

[5]The resulting formalism is similar to the Dotsenko-Fateev Coulomb gas construction [18], although the logic is different, Here the extra charges are forced on us by the physics, while in [18] one adds extra charges at infinity by hand and studies the structure of the resulting theory.

was accomplished in [19, 20].[6] However the computation can be performed rather easily in the long cylinder limit $T \gg L$, which is enough to extract the central charge and operator dimensions. In this limit the two boundaries of the cylinder don't talk to each other and so the factor (2.10) just gives some overall rescaling of the partition function. In addition we expect that the typical value of $\phi(T)$ will be large, and so we can treat the boundary condition $\phi(T)$ as a continuous variable rather than quantized. So we are led to evaluating the path integral

$$\int [D\phi] e^{-S+ie_0\phi(T)}, \tag{2.13}$$

with boundary conditions $\phi(0) = 0$, $\phi(T) = h = const$ and $S$ as in (2.6). We split $\phi$ into the classical component $\phi_{\text{cl}}$ and the fluctuation $\delta\phi$ satisfying the Dirichlet boundary conditions. We first integrate over $\delta\phi$, and then over $h$, the latter integral being

$$\int dh \, e^{-S_{\text{cl}}(h)+ie_0 h}, \qquad S_{\text{cl}}(h) = \frac{g}{4\pi}(h/T)^2 LT. \tag{2.14}$$

This gives an extra factor $e^{-(\pi e_0^2/g)(T/L)}$ in the partition function, which is interpreted as the reduction of the central charge $c$ from the free scalar boson value $c = 1$ to

$$c_{\text{Potts}} = 1 - \frac{6e_0^2}{g} \tag{2.15a}$$

$$= 1 - 6\frac{(2-g/2)^2}{g} = 13 - 6\left(\frac{g}{4} + \frac{4}{g}\right). \tag{2.15b}$$

Recall that the partition function should scale as $Z \sim e^{(\pi c/6)(T/L)}$ for $T \gg L$.

Let us proceed to discuss the operator spectrum. One interesting class of operators are the electric vertex operators $\mathcal{V}_e = e^{ie\phi}$. The set of allowed electric charges can be determined by the following argument. In the oriented loop model description, if we change orientation of a loop, the height inside will change by $2\phi_0 = \pi$. On the other hand in terms of clusters the loop orientation has no observable meaning. This means that any vertex operator playing a role in the Potts model should be invariant under such a change, i.e. $e \in 2\mathbb{Z}$.[7]

Scaling dimensions of these vertex operators can be predicted by a path integral argument based on (2.13). Namely, we insert an extra factor $e^{ie\phi(T)}$ and compute the partition function on a long cylinder. The change in the free energy compared to $e = 0$ gives the scaling dimension:

$$\Delta(\mathcal{V}_e) = \frac{1}{2g}((e_0 + e)^2 - e_0^2). \tag{2.16}$$

While Eqs. (2.15a) and (2.16) are standard for free scalar boson CFTs with background charge (see e.g. [21], section 9), we chose for completeness to include their direct derivation starting from (2.13) in the above review. It's also very important to emphasize that only some aspects of the Potts model can be understood from the Coulomb gas descriptions.

Main relations between parameters characterizing the $Q$-state Potts model are summarized in Table 1.

---

[6]On the contrary the partition function of the $F$-model on the cylinder is trivial to evaluate: one computes the gaussian path integral as a function of the height difference $\phi(T)$, and sums over $\phi(T) = n(\pi/2)$ [19]. Notice that it is legitimate to use the gaussian action (2.6) to compare relative weights of sectors not only for $\phi(T) \gg 1$ but also for $\phi(T) = O(1)$.

[7]Another argument which gives the same prediction is as follows. In the large volume limit, the distribution of $\phi$ at any point $x$ will become periodic with period $2\phi_0$ because of succession of many loops surrounding $x$ which can take any orientation. Correlation functions of vertex operators which are not invariant under $\phi \to \phi + 2\phi_0$ will thus vanish in the infinite volume limit.

Table 1: Main relations between parameters characterizing the $Q$-state Potts model. See section 2.3 for $u, g$, section 2.4 for $e_0, c_{\text{Potts}}$, and section 2.5 for $t, m$.

| Parameter | Meaning | Relation |
|---|---|---|
| $u$ | parameter of the oriented loop model | $\sqrt{Q} = 2\cos(4u)$ |
| $g$ | coupling of the gaussian height model (critical $2 < g \leqslant 4$, tricritical $4 \leqslant g < 6$) | $Q = 2 + 2\cos\frac{\pi g}{2}$ |
| $e_0$ | Coulomb gas charge | $e_0 = 2 - g/2$ |
| $c_{\text{Potts}}$ | CFT central charge | Eq. (2.15) |
| $t$ | parameter in Eq. (2.31) for Kac-degenerate dimensions | $t = \max(g/4, 4/g) \geqslant 1$ |
| $m$ | index of the minimal model $\mathcal{M}_m$ with the same central charge | $t = (m+1)/m$ |

## 2.5 Torus partition function

The full partition function $\mathbf{Z}_Q$ of the $Q$-state Potts model on the torus was found in the classic paper [3]. Let us describe the result and how it was obtained. The basic building block is the partition function of the free boson (2.6) with frustrated boundary conditions around the two cycles of the torus:

$$Z_{m,m'}(g) = \int_{\delta_1\phi=2\pi m,\, \delta_2\phi=2\pi m'} [D\phi]e^{-S}. \tag{2.17}$$

Summing these over frustration multiples of $f$, one defines the partition function of the compactified boson with compactification radius $f$:

$$Z_c[g,f] = f \sum_{m,m'\in f\mathbb{Z}} Z_{m,m'}(g). \tag{2.18}$$

This quantity has expansion in terms of the usual torus modulus $q = e^{2\pi i\tau}$ and $\bar{q}$:

$$Z_c[g,f] = \frac{1}{\eta(q)\eta(\bar{q})} \sum_{\substack{e\in\mathbb{Z}/f \\ m\in\mathbb{Z}f}} q^{x_{em}}\bar{q}^{\bar{x}_{em}}, \tag{2.19}$$

where $\eta(q) = q^{\frac{1}{24}}\mathcal{P}(q)$ is the Dedekind eta function, $\mathcal{P}(q) = \prod_{N=1}^{\infty}(1-q^N)$. The weights $x_{em}$, $\bar{x}_{em}$ are labeled by electric and magnetic charges $e$ and $m$:

$$x_{em}, \bar{x}_{em} = \frac{1}{4}(e/\sqrt{g} \pm m\sqrt{g})^2, \tag{2.20}$$

so that the scaling dimension and spin of the corresponding operators are given by:

$$x_{em} + \bar{x}_{em} = \frac{e^2}{2g} + \frac{g}{2}m^2, \quad x_{em} - \bar{x}_{em} = em. \tag{2.21}$$

We also need a modification of the compactified partition function (2.18) given by the following equation:

$$\hat{Z}[g,e_0] = \sum_{M',M\in\mathbb{Z}} Z_{M'/2,M/2}(g)\cos(\pi e_0 M' \wedge M), \tag{2.22}$$

where $a \wedge b = \gcd(a, b)$ is the greatest common divisor, with $a \wedge 0 = a$ by convention, introduced for the following reason. As for the cylinder case above, the $F$-model (which renormalizes to the free scalar boson) does not correctly reproduce the weights of noncontractible loop. The factor $\cos(\pi e_0 M' \wedge M)$ corrects for this mismatch, analogously to inserting the charges $\pm e_0$ at the ends of the cylinder in section 2.4.

The so defined $\hat{Z}[g, e_0]$ can be expanded in series in $q$ and $\bar{q}$ [3]:

$$\hat{Z}[g, e_0] = \frac{1}{\eta(q)\eta(\bar{q})} \left[ \sum_{P \in \mathbb{Z}} (q\bar{q})^{x_{e_0+2P,0}} + \sum_{\substack{M,N=1 \\ N \text{ divides } M}}^{\infty} \Lambda(M, N) \sum_{\substack{P \in \mathbb{Z} \\ P \wedge N = 1}} q^{x_{2P/N,M/2}} \bar{q}^{\bar{x}_{2P/N,M/2}} \right]. \quad (2.23)$$

Coefficients $\Lambda(M, N)$ are given by Eq. (3.24) of [3]. In general they depend on the factorization of $M$ and $N$ into prime integers. Below we will only need the following two partial cases for $M = 1$ or $M > 1$ prime:

$$\Lambda(M, 1) = 2 \left( \frac{\cos(\pi e_0 M) - \cos(\pi e_0)}{M} + \cos \pi e_0 \right), \quad (2.24)$$

$$\Lambda(M, M) = 2 \left( \frac{\cos(\pi e_0 M) - \cos(\pi e_0)}{M} \right) \quad (M \neq 1). \quad (2.25)$$

$\Lambda(M, N)$ are polynomials in $Q$ with rational coefficients, which implies that multiplicities $M_{h,\bar{h}}$ will also be polynomial in $Q$.

In terms of the defined objects, the torus partition function of the Potts model takes the form:[8]

$$\mathbf{Z}_Q = \hat{Z}[g, e_0] + \frac{1}{2}(Q - 1)(Z_c[g, 1] - Z_c[g, 1/2]). \quad (2.26)$$

Basically, $\hat{Z}[g, e_0]$ provides most of the answer, while the second term corrects a mismatch between the Potts model and the loop model for clusters having the cross topology, see [3] for details. We recall that $g$ is related to $Q$ by (2.7), choosing $2 < g \leqslant 4$ ($4 \leqslant g < 6$) for the critical (tricritical) case, and $e_0$ is given by (2.12).

As mentioned, the $Q = 2, 3$ critical and tricritical Potts theories are unitary minimal models, while $Q = 4$ can be described as an orbifold theory, as we review briefly in appendix C. In all these cases partition function can be computed independently and the result agrees with (2.26) [3].

## 2.6 Spectrum of primaries for $Q \leqslant 4$

We will now use the torus partition function to discuss the spectrum of primaries of the critical and tricritical Potts model. Recall that the torus partition function of any CFT can be written as

$$Z = (q\bar{q})^{-\frac{c}{24}} \operatorname{Tr} q^{L_0} \bar{q}^{\bar{L}_0}, \quad (2.27)$$

with Hamiltonian $H = L_0 + \bar{L}_0 - c/12$ and momentum $P = L_0 - \bar{L}_0$. Expanding $Z$ in $q, \bar{q}$ one can read off the spectrum of all states present in the theory and their multiplicities:

$$Z = (q\bar{q})^{-\frac{c}{24}} \sum_{h,\bar{h} \in \text{all}} M_{h\bar{h}} q^h \bar{q}^{\bar{h}}. \quad (2.28)$$

---

[8]Notice that one should not confuse electric and magnetic charges $e, m$ of the states in this expression with the Coulomb gas charge of operator $\mathcal{V}_e$ in (2.16). In particular the constraint $e \in 2\mathbb{Z}$, does not apply to $e$.

Furthermore, the partition function is also expandable into Virasoro characters $\chi_h$ of primaries:

$$Z(q,\bar{q}) = (q\bar{q})^{-\frac{c}{24}} \sum_{h,\bar{h} \in \text{primaries}} N_{h\bar{h}} \, \chi_h(q)\chi_{\bar{h}}(\bar{q}), \qquad (2.29)$$

the sum being over the weights $h$, $\bar{h}$ of the primaries, with $N_{h,\bar{h}}$ their multiplicities. On the other hand the sum in (2.28) is over both primaries and descendants, so that sum contains many more terms, and with different multiplicities.

Below we will encounter two types of Virasoro characters. First, for generic $h$ (non-degenerate primary), the character is given by[9]

$$\chi_h(q) = q^h/\mathcal{P}(q). \qquad (2.30)$$

Second, there will be cases when $h$ is a Kac degenerate representation $h = h_{r,s}$. For the central charge as in (2.15) these are given by (see [21], Eq. (7.31))

$$h_{r,s} = \frac{t}{4}(r^2 - 1) + \frac{1}{4t}(s^2 - 1) + \frac{1 - rs}{2}, \qquad t = \max(g/4, 4/g), \qquad (2.31)$$

with $r,s$ positive integers.[10,11] The representation is then degenerate at level $rs$, the null descendant having weight $h_{r,-s}$. Below we will mostly focus on the generic $Q$ case, when this descendant itself is not degenerate.[12] In this case the character of the $h_{r,s}$ primary is given by:

$$\chi_{h_{r,s}}(q) = (q^{h_{r,s}} - q^{h_{r,-s}})/\mathcal{P}(q). \qquad (2.33)$$

A particular case of (2.33) occurs for the unit operator: $h = 0 = h_{1,1}$, when

$$\chi_0(q) = (1 - q)/\mathcal{P}(q). \qquad (2.34)$$

From (2.26), (2.19), (2.23) we get expansion (2.28) of the Potts model partition function. Notice, however, that 'multiplicities' $M_{h,\bar{h}}$ are given by polynomials in $Q$, so they are in general not integer, unless $Q$ is integer.[13] This may sound puzzling since normally multiplicities are integer. To understand the origin of this subtlety, let us go back to the lattice partition function from section 2.1. In the continuum limit, the factors $q^h \bar{q}^{\bar{h}}$ in (2.28) originate from the eigenvalue factors $\lambda^N$ on the lattice in (2.3), while the multiplicities $M_{h,\bar{h}}$ are the weights $M_\lambda$ in (2.3). As discussed in section 2.1 these weights for non-integer $Q$ do not just count the eigenvalues, but involve a matrix element of a gluing operator, which explains why they don't have to be integers nor even positive.[14]

---

[9]We don't include the factor $q^{-c/24}$ into the character.

[10]The two branches $t = g/4$ and $t = 4/g$ are related by interchanging $r$ and $s$. By choosing always $t > 1$ we ensure that for $t = (m+1)/m$ the weight numbering will agree with the form

$$h_{r,s} = \frac{[(m+1)r - ms]^2 - 1}{4m(m+1)}, \qquad (2.32)$$

conventional in the unitary minimal models. This will be convenient in section 4.

[11]Notice that not all operators appearing in partition function are Kac-degenerate. In some literature on the Potts model weights of non-degenerate operators are also represented as $h_{r,s}$ with $r,s$ non-integer. This will not be done in our work, where notation $h_{r,s}$ will be used only with $r,s$ integer.

[12]On the other hand, in minimal models $\mathcal{M}_{p,p'}$ we have $h_{r,-s} = h_{p'+r,p-s}$ and so it may be degenerate. The character then takes a more complicated form than (2.33).

[13]These multiplicities have also been studied in [22].

[14]Note that negativity of some weights is not a direct consequence of the model being non-unitary. For example, while the Lee-Yang model is non-unitary, its torus partition function decomposes into characters with positive integer weights.

Up to the prefactor $1/[\eta(q)\eta(\bar{q})]$, the obtained expansion of $\mathbf{Z}_Q$ consists of terms of the form $q^{x_{em}}\bar{q}^{\bar{x}_{em}}$, times multiplicities. We will introduce notation $\mathcal{O}_{e,m}$ for the state corresponding to such a term. Its conformal weight is given by

$$h = x_{em} + \frac{c-1}{24}, \qquad \bar{h} = \bar{x}_{em} + \frac{c-1}{24}. \tag{2.35}$$

First, we have states with $e = e_0 + 2P$ ($P \in \mathbb{Z}$) and $m = 0$, coming from the first term in $\hat{Z}[g, e_0]$. Notice that $\mathcal{O}_{e_0+2P,0} = \mathcal{V}_{2P}$, the vertex operator of the same dimension introduced in the Coulomb gas description. This identification is not accidental, the Coulomb gas scaling dimension (2.16) and the term $\propto (q\bar{q})^{x_{e_0+2P,0}}$ in $\mathbf{Z}_Q$ having essentially the same path-integral origin.

Second, we have states with $e$ and $m$ both rational numbers, coming from $Z_c[g, f]$ and from the second term in $\hat{Z}[g, e_0]$. We should identify $\mathcal{O}_{e,m} \equiv \mathcal{O}_{-e,-m}$ since they have the same conformal weights. These states do not have an obvious Coulomb gas interpretation.

In this notation, $\mathbf{Z}_Q$ takes the form of the sum of nondegenerate characters (2.30) of states $\mathcal{O}_{e,m}$. Does this mean they are all primaries? Not so fast. We have to watch out that some $\mathcal{O}_{e,m}$ are Kac-degenerate. More work is then needed to reorganize the decomposition in terms of Virasoro characters, during which process some states may drop out from the list of primaries. We also need to check for more banal *spectrum coincidences*, which may lead to total cancellation of multiplicities. To study these degeneracies we write $h, \bar{h}$ as $r_1 g^{-1} + r_2 g + r_3$ with $r_i$ rational:

$$\mathcal{V}_{2P} \equiv \mathcal{O}_{e_0+2P,0}: \quad h = \bar{h} = P(P+2)g^{-1} - P/2\,,$$
$$\mathcal{O}_{e,m}: \quad h, \bar{h} = \left(\frac{e^2}{4}-1\right)g^{-1} + \left(\frac{m^2}{4}-\frac{1}{16}\right)g + \frac{1}{2} \pm em/2\,. \tag{2.36}$$

Apart from $\mathcal{O}_{e,m} \equiv \mathcal{O}_{-e,-m}$ mentioned above, there is only one spectrum coincidence which holds for any $Q$: it is between $\mathcal{V}_{-2}$ and the operators $\mathcal{O}_{0,\pm 1/2}$. The total multiplicity comes out $-(Q-1) + 2\cos\pi e_0 + 1 = 0$, so these states do not actually exist.[15]

The first Kac degeneracy occurs for the operator $\mathcal{V}_0$. It has dimension 0 and is identified with the unit operator; clearly it is Kac-degenerate. The full unit operator character is (see Eq. (2.34))

$$\chi_0(q)\chi_0(\bar{q}) = (1-q)(1-\bar{q})/[\mathcal{P}(q)\mathcal{P}(\bar{q})]. \tag{2.37}$$

We have to see how this character emerges from combining various nondegenerate characters. Expanding the numerator, the first term $1/[\mathcal{P}(q)\mathcal{P}(\bar{q})]$ is precisely the contribution of $\mathcal{V}_0$. One missing term, $q\bar{q}/[\mathcal{P}(q)\mathcal{P}(\bar{q})]$, comes from $\mathcal{V}_{-4}$, of scaling dimension 2, which as a consequence does not exist as a primary operator.[16]

The other missing term in the unit operator character is the cross term $(-q-\bar{q})/[\mathcal{P}(q)\mathcal{P}(\bar{q})]$. This comes from the operators $\mathcal{O}_{e,m}$ with $e = \pm 2$, $m = \pm 1/2$ which have spin 1 and dimension 1 for any $g$. These operators appear in the second term in (2.23) ($M = N = P = 1$) as well as

---

[15]We exclude as contrived the possibility that a state exists but does not contribute to the partition function for any $Q$.

[16]In fact, even more is true. Were we to expand the partition function in powers of $q, \bar{q}$ (including the Dedekind factors), we would see that the theory does not contain *any* scalar of dimension 2, primary or descendant. This is in contradiction with the discussion of Kondev [23], who argued that operator $\mathcal{V}_{-4} = e^{-i4h}$ must be 'exactly marginal', and used this to determine the relation between $Q$ and $g$, instead of borrowing the $F$-model result (2.7). While the calculation behind Kondev's argument is correct, physical interpretation should be changed because as we showed the operator he calls 'exactly marginal' does not even exist. We propose a different interpretation in appendix A.

in $Z_c[g, 1/2]$. Their total coefficient is $2\cos(\pi e_0) - (Q-1) = -1$ as needed. The conclusion is that we reproduce correctly the character of the unit operator, while operators $\mathcal{V}_{-4}$ and $\mathcal{O}_{\pm 2, \pm 1/2}$ drop out from the spectrum of primaries.

We will now carry out similar analysis for a few more prominent low-lying primary operators. We will see more examples of Kac-degenerate states, and degenerate characters arising as sums of non-degenerate ones. We will not, however, attempt to rewrite the full partition function in terms of Virasoro characters.

### Singlets

The operators $\mathcal{V}_{2P}$ occur in the first term of $\hat{Z}[g, e_0]$ with multiplicity 1 for any $Q$, and therefore we will refer to them as 'singlets'. As discussed above the case $P = 0$ corresponds to the unit operator, and $P = -1, -2$ do not exist. The lightest non-unit operators are for $P = 1, 2$, which we call the energy operator $\varepsilon$ and the subleading energy operator $\varepsilon'$. On the tricritical branch both $\varepsilon$ and $\varepsilon'$ are relevant, while on the critical branch only $\varepsilon$ is relevant. When the two branches meet we have $\Delta_{\varepsilon'} = 2$.

Comparing (2.36) with (2.31), we see that all $\mathcal{V}_{2P}$, $P \geqslant 0$, are Kac-degenerate. In fact $h = h_{1, P+1}$ on the tricritical or $h = h_{P+1, 1}$ on the critical branch. One can show that just like for the unit operator the terms in $\mathbf{Z}_Q$ recombine nicely to give the degenerate characters $\chi_{h_{r,s}}(q)\chi_{h_{r,s}}(\bar{q})$. The scalar term $\propto q^{h_{r,-s}}\bar{q}^{h_{r,-s}}$ comes from $\mathcal{V}_{2P'}$ with $P' = -P - 2$, whose multiplicity is also 1. The spin-1 terms $-q^{h_{r,s}}\bar{q}^{h_{r,-s}} - q^{h_{r,-s}}\bar{q}^{h_{r,s}}$ come from $\mathcal{O}_{e,m}$ with $e = \pm 2(P+1)$ and $\pm m = 1/2$, with precisely the right coefficient. The ability to rewrite the partition function in this form means that the operators $\mathcal{V}_{2P'}$ and $\mathcal{O}_{\pm 2(P+1), \pm 1/2}$ do not appear in the spectrum of primaries.[17]

As mentioned the operators $\mathcal{V}_{2P} \equiv \mathcal{O}_{e_0 + 2P, 0}$ should be identified with electric vertex operators $e^{ie\phi}$ with $e = 2P$ whose dimension was computed using Coulomb gas method in (2.16). The above discussion establishes that for $P \geqslant 0$ these vertex operators are primaries.

As far as $\mathcal{V}_{2P'}$ with negative charge $P' \leqslant -1$, the above discussion is summarized as follows. For $P' = -1$ this operator simply does not exist since multiplicity is exactly zero. For $P' \leqslant -3$ a scalar operator of such dimension exists, but is interpreted as not a primary but as a descendant of the positive charge primary $\mathcal{V}_{2P}$, $P = -P' - 2$, at level $(P+1, P+1)$. The latter reasoning formally applies also for $P' = -2$, but since the unit operator has no descendants at the level $(1, 1)$, this means that $\mathcal{V}_{-4}$ does not exist. This may be a surprise from the point of view of the Coulomb gas construction, but that's what the torus partition function tells us.[18]

### Vectors

We will call 'vectors' operators of multiplicity $Q - 1$, the dimension of the lowest nontrivial representation of $S_Q$. The lowest such operators are $\mathcal{O}_{e,0}$ with $e = \pm 1, \pm 3$, which come from $Z_c[g, 1]$. These are the spin and the subleading spin operators $\sigma$ and $\sigma'$. For generic $Q$ these operators are non-degenerate primaries.

---

[17]One might contemplate the possibility that primaries of such dimension actually secretly exist, but their contribution to the torus partition function, as well as to the annulus partition function [20], is exactly zero for all $Q$. Perhaps these may be null descendants of Kac-degenerate primaries, not necessarily identically vanishing in non-unitary theories. We discard such a possibility as unduly contrived.

[18]Additional evidence for the absence of these operators can be obtained from the cylinder partition function [20], Eq. (2). Organizing it in characters, one sees that negative electric charge operators are absent from the list of primaries.

**Higher representations**

An interesting light operator is $\mathcal{O}_{0,1}$, from the $P = 0$, $M = 2$, $N = 1$ term in the second sum of $\hat{Z}[g, e_0]$. Using (2.25), its multiplicity is

$$\cos \pi e_0 + \cos 2\pi e_0 = \frac{Q(Q-3)}{2}. \tag{2.38}$$

This matches the dimension of an $S_Q$ representation given by the Young tableau with two rows with $Q - 2$ and 2 boxes in them, which for integer $Q$ exists only for $Q \geqslant 4$. Notice that this multiplicity becomes negative for $Q < 3$. Again, for generic $Q$ this operator is a non-degenerate primary.

See appendix B for operators corresponding to even larger Young tableaux. For any Young tableau $Y$, the corresponding representation exists for all sufficiently large $Q$ and has dimension $D_Y(Q)$ which is a polynomial in $Q$. One can take this polynomial and analytically continue it to $Q \leqslant 4$. In all cases that we checked, multiplicities of operators predicted by (2.26) are decomposable into sums of $D_Y(Q)$ with coefficients which are positive $Q$-independent integers (appendix B).[19] It is not fully obvious why this should be the case. Of course for integer $Q$ we have true $S_Q$ symmetry and so the decomposition of multiplicities in dimensions of irreps of $S_Q$ should be possible for each individual $Q$. The nontrivial part is that such a decomposition extends to non-integer $Q$ with $Q$-independent coefficients. This may be related to the fact that the Deligne category mentioned in section 2.2 is a semisimple tensor category.[20]

In the above discussion we focused on generic $Q$. For $Q = 2, 3, 4$ the discussion needs to be modified. First of all the theory has a local spin description, so all multiplicities must be positive integers. In addition, for $Q = 2, 3$, the theories are minimal models and contain a finite number of primary fields. (On the contrary, for generic $Q$ we have infinitely many primaries.) Various terms in the partition function must then recombine, to either cancel or form characters of degenerate primaries in the minimal models, which are more complicated than (2.33). Similar complicated cancellations have to happen for an infinite set of $Q$'s which give a rational central charge $c = c_m = 1 - 6/[(m+1)m]$ corresponding to the unitary minimal models, even though the spectrum of the Potts model will have only partial overlap with the corresponding minimal model unless $Q = 2, 3$ (see section 4).

A very simple example of such a cancellations happens for the operator with multiplicity (2.38). For $Q = 3$ this operator simply disappears. For $Q = 2$, its multiplicity is negative. However, for $Q = 2$ this operator is degenerate with the vector operator $\mathcal{O}_{\pm 3,0}$ along the critical branch and with $\mathcal{O}_{\pm 5,0}$ for the tricritical branch. The total multiplicity is zero: this operator does not exist for $Q = 2$.

The spectrum of several light scalar operators and their multiplicities are summarized on Fig. 7.

# 3 Analytic continuation to $Q > 4$

The 2d Potts model at criticality is normally discussed only for $Q \leqslant 4$. For $Q > 4$, the phase transition is first-order, and one does not expect to find any fixed points. As far as real fixed

---

[19]Ref. [3] contains a tangential remark to the contrary for the analogous case of the $O(n)$ model, which appears to be an error. In any case, this minor discrepancy does not affect any of their other conclusions. We thank Hubert Saleur for a discussion.

[20]We are grateful to Damon Binder for very useful discussions concerning this point, which will be elaborated elsewhere.

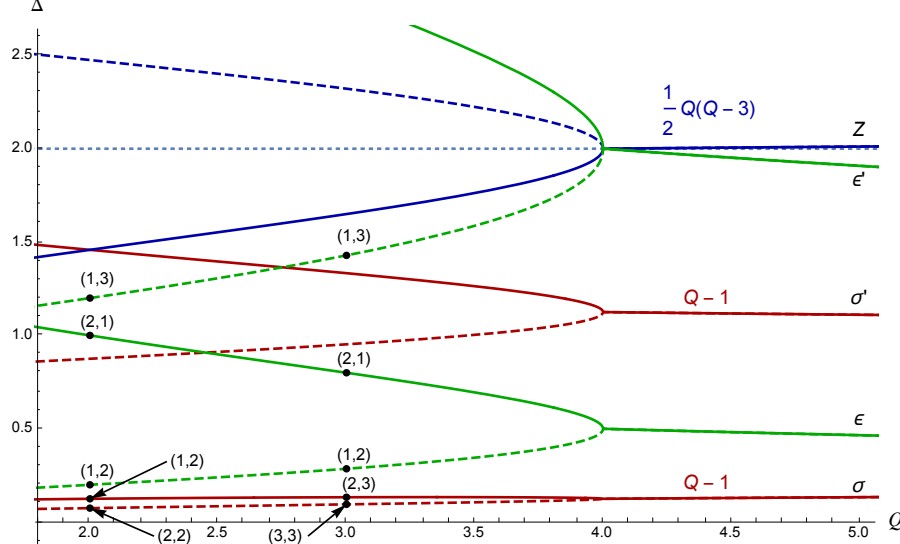

Figure 7: Dimensions of light scalar operators as functions of $Q$. The $Q \leqslant 4$ region corresponds to the critical (solid) and tricritical (dashed) Potts models. Singlet operators are in green, other multiplicities are marked. $\Delta = 2$ is the marginality line. The $(r, s)$ positions in the Kac table are shown for $\varepsilon$, $\varepsilon'$, as well as for $\sigma$ which is degenerate for $Q = 2, 3$. Notice that $\varepsilon$ and $\varepsilon'$ remain Kac-degenerate with the same integer $(r, s)$ also for non-integer $Q$, while this is not true for the other shown operators. For $Q > 4$ we show the real parts of the analytically continued dimensions, see section 3.

points are concerned this expectation is correct.[21] However, instead there exists a pair of complex fixed points. In the previous section we saw that critical and tricritical Potts models become identical at $Q = 4$. Following the discussion in [1] it is convenient to visualize these fixed points as CFTs moving in the theory space as the parameter $Q$ is varied, see Fig. 8. The fixed points have the same symmetries and annihilate at $Q = 4$. Ref. [1] reviewed the abundant evidence from prior work [25–28] that in the vicinity of this point, the beta-function controlling the RG flow is of the form (1.1) with $y \sim Q - 4$, suggesting the existence of two fixed points at couplings $\lambda \sim \pm i \sqrt{Q - 4}$.

In [1] we discussed some examples of perturbative complex theories defined by Lagrangians with complex coupling constants. The Potts model is a strongly coupled theory and consequently we don't expect to find any perturbative description. The $Q > 4$ complex Potts CFTs can in principle be constructed with the help of lattice models reviewed in section 2.1, either the spin or the cluster one, if one complexifies coupling constants in these models and tunes them to the critical values. To find these fixed points on the lattice, one will need to tune two complex couplings, which can be taken e.g. the temperature and the vacancy fugacity in the diluted Potts model (see footnote 27 below).

Here we will pursue an alternative method to explore the $Q > 4$ CFTs — by means of analytic continuation from $Q \leqslant 4$. In the previous section we discussed operator dimensions and their multiplicities in the torus partition function, which for $Q \leqslant 4$ are explicit analytic functions of $Q$. It is reasonable to assume that observables in the complex CFTs at $Q > 4$

---

[21]In [24], some $S_Q$ invariant CFTs with $4 < Q \lesssim 5.56$ were predicted to exist using the massless scattering theory method. Their theories are real and presumably can be realized as continuous phase transitions in the antiferromagnetic Potts model. There is no relation to the theories considered here. We thank Gesualdo Delfino for discussions.

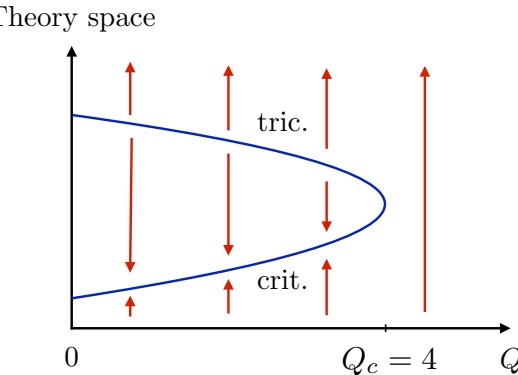

Figure 8: Annihilation of critical and tricritical fixed points.

will agree with the analytic continuation of the corresponding observable form $Q \leqslant 4$. In this paper we will only consider analytic continuation in the neighborhood of the real $Q$ axis. We will perform a set of consistency checks that complex CFTs at real $Q > 4$ defined by means of such analytic continuation indeed describe the complex fixed points of the Potts model. In the future it would be interesting to explore analytically continued CFTs far away from the real $Q$ axis, and to understand their physical meaning if any.

Analytic continuation is easiest for the central charge and the scaling dimensions, since these can be expressed as rational functions of $g$, Eqs. (2.15), (2.36). We will keep these expressions, but we will analytically continue $g(Q)$, given by Eq. (2.8), from $Q \leqslant 4$ to $Q > 4$.

For $Q \to 4^-$ we have $g(Q) \approx 4 \pm (2/\pi)\sqrt{4-Q}$. For $Q > 4$, $g(Q)$ develops a branch cut singularity, $g(Q) \approx 4 \pm i(2/\pi)\sqrt{Q-4}$. The exact expression is

$$g(Q) = 4 \pm i\frac{2}{\pi} \log \frac{Q-2+\sqrt{Q(Q-4)}}{2} \qquad (Q > 4). \tag{3.1}$$

Notice that the real part of $g$ remains constant. We will call $\mathcal{C}$ the complex CFT corresponding to the $+$ branch, while the other branch is $\overline{\mathcal{C}}$.

Equivalently, we perform analytic continuation from the tricritical branch going around the branch point $Q = 4$ from below in the complex $Q$ plane, so that $\sqrt{4-Q}$ for $Q < 4$ goes to $i\sqrt{Q-4}$ for $Q > 4$, where both square roots are positive. The choice between analytic continuation from the tricritical and critical branch is arbitrary[22] — continuing from the critical branch with the same contour, or from the tricritical branch in the opposite direction, would get $\overline{\mathcal{C}}$ instead of $\mathcal{C}$. That is, the real parts of all scaling dimensions would be the same, and the imaginary parts would change signs. This is why there are two CFTs for $Q > 4$, one being the complex conjugate of the other. Finally, if one goes around the $Q = 4$ point in the complex plane and comes back to real $Q < 4$ the tricritical theory becomes the critical one and vice versa.

The real parts of analytically continued dimensions of a few low-dimension operators can be read off from the right side of Fig. 7. Analogously to the dimensions, the central charge (2.15) of the Potts CFT develops an imaginary part for $Q > 4$, the central charges of $\overline{\mathcal{C}}$ and $\mathcal{C}$ being complex conjugate.

---

[22]It is somewhat more convenient to continue from the tricritical branch, as it contains the relevant operator $\varepsilon' = \phi_{1,3}$, crucial for our computations below, which drives the RG flow from the tricritical to the critical branch. On the critical branch we have $\varepsilon' = \phi_{3,1}$ which is irrelevant.

We see that the $Q > 4$ Potts CFTs contain operators with complex scaling dimensions, and the corresponding conjugate operators are not present in the same theory. In terminology of [1], this means that these CFTs are complex, as opposed to real.[23] Moreover there are two complex theories with opposite imaginary parts of all observables, and the size of these imaginary parts is controlled by the parameter $\sqrt{Q-4}$. In such a situation, the real RG flow squeezed between the two complex CFTs exhibits the walking behavior [1].

Turning next to the multiplicities of primary operators $M_{h,\bar{h}}$ (see section 2.6), they are polynomials in $Q$ and their analytic continuation is straightforward. In particular, they stay real for any real $Q$. As mentioned, it appears that $M_{h,\bar{h}}$ can be represented as a sum of dimensions of irreducible representations of $S_Q$ analytically continued in $Q$, with $Q$-independent positive integer coefficients. In particular, when we specialize to integer $Q \geqslant 5$, we expect complex Potts CFTs to have a true $S_Q$ symmetry. For example, there should exist a pair of $S_5$-symmetric complex CFTs of central charge

$$c = 13 - 6\left(\frac{g(5)}{4} + \frac{4}{g(5)}\right) \approx 1.138 \pm 0.021 i \,. \tag{3.2}$$

To summarize, we started with the analytic expression for the torus partition function of the critical Potts model with real $Q < 4$ and analytically continued operator dimensions and multiplicities to $Q > 4$. This is equivalent to the analytic continuation of the partition function itself. In particular all the partition function properties for $Q < 4$, like modular invariance, continue to hold for $Q > 4$. The information extracted from the partition function is consistent with our conjecture about the existence of complex fixed points at $Q > 4$. Further checks involving OPE coefficients will be given below.

## 4 Walking RG flow in $Q > 4$ Potts models

We will now use the knowledge of the complex CFTs $\mathcal{C}$ and $\bar{\mathcal{C}}$ to study the walking RG flow in the Potts model for $Q \gtrsim 4$. Basic framework of how to do this was presented in [1], section 6.3. The real flow trajectory passes halfway between the two complex CFTs. The part of the trajectory close to the CFTs exhibits approximately scale-invariant behavior, and can be accessed using a form of conformal perturbation theory (CPT). Here we will demonstrate this framework by concrete computations.

CPT computations require operator dimensions in our complex CFTs, known from the partition function. We will also need OPE coefficients as well as some integrals of the 4pt functions (see [29] for a review of first-order and [30–32] for second-order CPT). In practice we will only need to know those up to some fixed order in $Q - 4$ since, as we will see momentarily, the imaginary part of the coupling constant in the walking region will itself be proportional to $Q - 4$. We would like to stress, however, that conceptually our procedure is quite different from expanding around the $Q = 4$ fixed point. One may imagine that conformal data at the complex fixed point is known exactly, and we are expanding only in the coupling constant of perturbation around this fixed point. In this sense our expansion is a usual CPT, albeit as we will see with a complex coupling. Instead expansion in $Q - 4$ around the $Q = 4$ Potts model is

---

[23]We refer the reader to [1] for a detailed definition of complex field theories. Here we simply note that the presence of complex scaling dimensions by itself does not yet imply that the theory is complex, in fact there exist real non-unitary CFTs in which operators have complex dimensions, however those operators always come in conjugate pairs.

on a less obvious footing since $Q-4$ itself isn't a coupling constant, but merely a parameter. In particular, models with different $Q$ have different symmetries and expansion around $Q=4$ requires deforming the symmetry, a feature that we would like to avoid.

Although the Potts models at general $Q$ are not exactly solved apart from their spectrum, some additional conformal data needed for CPT can be determined for arbitrary $Q$ for the following reason. Comparing (2.31) and (2.36), some of the light fields present in the Potts model belong to degenerate conformal families for any $Q$, and consequently their correlation functions satisfy certain differential equations [33]. In principle, this fixes the correlators up to a few constants that can be determined by requiring proper crossing symmetry properties, although in practice this procedure is still rather complicated.

There is, however, another helpful trick. For a discrete infinite set of $Q$'s between 2 and 4 (called the Beraha numbers) the Potts model central charge agrees with that of the diagonal minimal models $\mathcal{M}_m \equiv \mathcal{M}(m+1,m)$, $c_m = 1-6/[m(m+1)]$.[24] We have (see Fig. 9)

$$\text{Tricritical:} \qquad Q = 2 + 2\cos\frac{2\pi}{m}, \qquad m = \frac{2\pi}{\arccos\left(\frac{Q}{2}-1\right)}, \qquad (4.1)$$

$$\text{Critical:} \qquad Q = 2 + 2\cos\frac{2\pi}{m+1}, \qquad m = \frac{2\pi}{\arccos\left(\frac{Q}{2}-1\right)} - 1. \qquad (4.2)$$

What is the significance of this agreement? As is well known, the tricritical and critical $Q=2$ Potts (i.e. Ising) are actually identical to $m=4$ and $m=3$, but for other $Q$'s for which $m$ is an integer, there is no exact coincidence. E.g. the $Q=3$ Potts models are nondiagonal minimal models – they contain only a subset of Kac-table operators, and with nontrivial multiplicities, as well as primaries with spin.

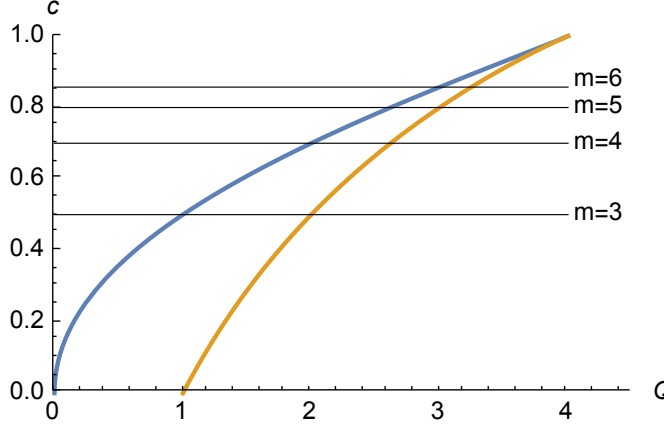

Figure 9: Central charge of the critical (yellow) and tricritical (blue) Potts model as a function of $Q$. The central charge of several unitary minimal models is indicated by horizontal lines.

Still, there are some operators, singlets under $S_Q$, which occur in all minimal models and in all Potts models. It seems reasonable to assume that their correlators should agree in both models. The reason is that these correlators satisfy both the differential equations and crossing relations, which for singlet operators are identical in Potts and in $\mathcal{M}_m$, and these conditions are extremely constraining. We can then use the OPE coefficients and 4pt functions

---

[24]Recall that the diagonal (or $A$-series) minimal models only contain scalar primaries $h = \bar{h} = h_{r,s}$ with multiplicity 1.

in the diagonal minimal models computed in [18, 34] (see also [35] and [36] for some more compact expressions) to learn about the Potts model.[25] Minimal models provide the OPE data as functions of $m$, which can be expressed via $Q$ through (4.1) and analytically continued.

Additional source of information is $Q = 4$, since the 4-state Potts model is known to be described by the free boson compactified on $S_1/\mathbb{Z}_2$ [41], and direct computations are possible. To guard against possible subtleties and to obtain information about operators not present in the minimal models, we crosscheck some of the results at $Q = 4$ (appendix C). In fact for some of our considerations knowing the values of OPE coefficients at $Q = 4$ will be just enough, while for others higher orders in $Q - 4$ are needed, requiring the analytic continuation from the minimal models.

## 4.1 One-loop beta-function

Based on the agreement of dimensions noticed in section 2.6, we have identification [18]:

$$\text{Critical: } \varepsilon = \mathcal{O}_{e_0+2,0} = \phi_{2,1}, \tag{4.3}$$

$$\varepsilon' = \mathcal{O}_{e_0+4,0} = \phi_{3,1}, \tag{4.4}$$

$$\text{Tricritical: } \varepsilon = \mathcal{O}_{e_0+2,0} = \phi_{1,2}, \tag{4.5}$$

$$\varepsilon' = \mathcal{O}_{e_0+4,0} = \phi_{1,3}, \tag{4.6}$$

as also indicated in Fig. 7. We are most interested in $\varepsilon'$, the subleading energy operator. This singlet operator is irrelevant at the critical point and relevant at the tricritical point for $Q < 4$. Operator $\phi_{1,3}$ is known to produce the RG flow between consecutive minimal models [42]. We therefore make an extremely plausible assumption that operator $\varepsilon'$ drives the flow from the tricritical Potts to the critical Potts for any $Q < 4$.[26] For $Q = 4$ the two CFTs collide, and $\varepsilon'$ becomes marginal.

Now consider analytic continuation to $Q > 4$. The dimension of $\varepsilon'$

$$\Delta^{\mathcal{C}}_{\varepsilon'} = 16/g(Q) - 2, \tag{4.7}$$

acquires an imaginary part, negative in the complex theory $\mathcal{C}$ (it would be positive in $\overline{\mathcal{C}}$), as well as a negative corrections to the real part (see Fig. 10). Near $Q = 4$ we have

$$\Delta^{\mathcal{C}}_{\varepsilon'} = 2 - 2i\frac{\epsilon}{\pi} - \frac{\epsilon^2}{\pi^2} + \dots \qquad (\epsilon = \sqrt{Q-4}). \tag{4.8}$$

We consider a family of RG flows perturbing $\mathcal{C}$ by

$$g_0 \int d^2x\, \varepsilon'(x), \tag{4.9}$$

for various initial values of $g_0$.

**Note:** From now on $g_0$ and $g$ will refer to the bare and renormalized CPT coupling in expansion around $\mathcal{C}$, to agree with the notation in [1]. These couplings have nothing to do

---

[25]Although it should be possible to extract OPE data by directly studying non-diagonal minimal models, this is rather more complicated than for the diagonal case, see e.g. [37] for a general discussion and [38] for $Q = 3$ Potts; for an alternative method of computing the OPE coefficients see [39, 40].

[26]This is not fully obvious since as mentioned the spectrum of the Potts models only partly overlaps with that of the diagonal minimal models.

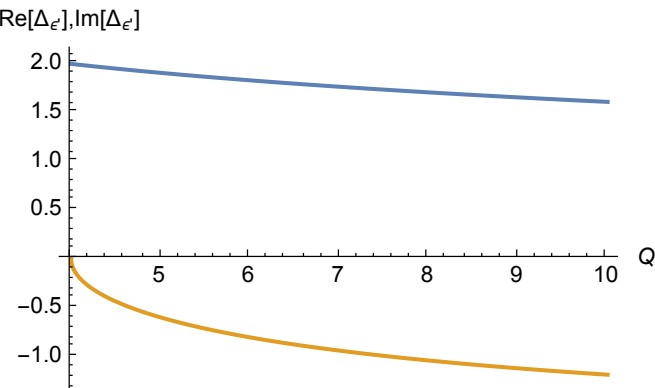

Figure 10: Real and imaginary part of $\Delta_{\varepsilon'}$ as a function of $Q$.

with the Coulomb gas coupling from section 2, which was denoted there $g$ or $g(Q)$. The latter coupling will not appear in the rest of the paper. Hopefully this will not create confusion.

The theory $\mathcal{C}$ also contains a strongly relevant singlet operator, analytic continuation of $\varepsilon$, whose coefficient is tuned to zero. We expect these flows to have topology shown in Fig. 11. Notice that because of the negative $O(\epsilon^2)$ correction to $\Delta_{\varepsilon'}^{\mathcal{C}}$ and $\Delta_{\varepsilon'}^{\overline{\mathcal{C}}}$, the trajectories starting at $\mathcal{C}$ or $\overline{\mathcal{C}}$ slowly unwind. This effect was not considered in [1], where the correction to the real part was neglected, leading to oversimplified flow topology shown in Fig. 2 of [1].[27]

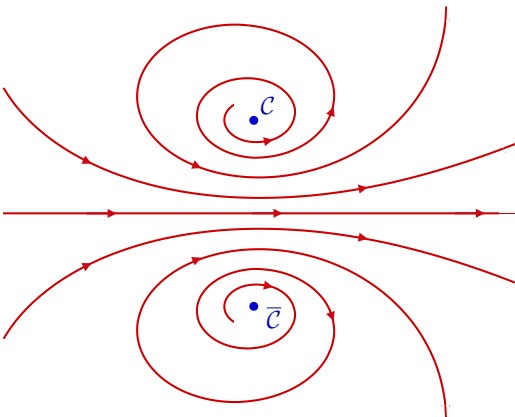

Figure 11: RG flow topology expected in the complexified Potts model at $Q > 4$.

The part of the RG trajectories close to $\mathcal{C}$ and $\overline{\mathcal{C}}$ can be studied in CPT. This concerns in particular the trajectory which does not even begin at $\mathcal{C}$ or $\overline{\mathcal{C}}$ but follows the horizontal line halfway between them. According to [1], this trajectory describes the walking RG flow.[28]

Let us recall how one can move between $\mathcal{C}$ and $\overline{\mathcal{C}}$, as well as study the walking RG trajectory, using leading order CPT ( [1], section 6.3). The one-loop beta-function reads [29]

$$\beta(g) = \varkappa g + \pi C_{\varepsilon'\varepsilon'\varepsilon'}g^2, \tag{4.10}$$

---

[27]Hence, to realize the complex fixed points $\mathcal{C}, \overline{\mathcal{C}}$ on the lattice, one has to finetune the couplings of both $\varepsilon$ and $\varepsilon'$. In total one needs to tune two complex, or four real couplings, making lattice studies of the complex fixed points in the Potts model somewhat nontrivial.

[28]In general the real RG flow doesn't have to follow an exact straight line, in fact this property is scheme dependent as we discuss in sections 4.4. Topology of the RG flow, however, doesn't depend upon the choice of scheme.

where $\varkappa = \Delta_{\varepsilon'}^{\mathcal{C}} - 2,$[29] and $g$ is the renormalized coupling constant which appears in the expressions for beta-functions and anomalous dimensions, as opposed to the bare coupling $g_0$ used in (4.9). At one loop we only need $\varkappa$ at $O(\epsilon)$ and $C_{\varepsilon'\varepsilon'\varepsilon'}$ at $O(1)$. For later uses we cite $C_{\varepsilon'\varepsilon'\varepsilon'}$ including the first subleading term:[30]

$$C_{\varepsilon'\varepsilon'\varepsilon'} = \frac{4}{\sqrt{3}} - \frac{2i\sqrt{3}}{\pi}\epsilon + O(\epsilon^2). \tag{4.11}$$

The beta-function vanishes at $g = 0$ and at

$$g = g_{FP} = -\varkappa/(\pi C_{\varepsilon'\varepsilon'\varepsilon'}) = i\frac{\sqrt{3}\epsilon}{2\pi^2} + \dots. \tag{4.12}$$

Notice that $g_{FP}$ is purely imaginary at the considered leading order. This second fixed point should correspond to $\overline{\mathcal{C}}$. The first check of this identification is the 'Im-flip': the imaginary part of dimension of $\varepsilon'$ changes sign when going from $\mathcal{C}$ to $\overline{\mathcal{C}}$. As shown in [1] this is completely general at this order, and we don't repeat the argument. However, we will see below several other checks of this identification, possible in the situation at hand.

Furthermore, the walking trajectory corresponds to $g$ of the form

$$g = g_{FP}/2 - \lambda, \tag{4.13}$$

with $\lambda$ real. Reality of $\lambda$ is preserved since in terms of $\lambda$ the beta-function is real [1]:

$$\beta_\lambda = -\beta(g_{FP}/2 - \lambda) = -\frac{\sqrt{3}\epsilon^2}{4\pi^3} - \frac{4\pi}{\sqrt{3}}\lambda^2. \tag{4.14}$$

For small $\epsilon$, this is the walking beta-function of the form (1.1) (up to rescaling $\lambda$). In particular, we can obtain the correlation length in the $Q > Q_c$ Potts models:

$$\xi_{\text{Potts}} = \exp\left(\int_{\lambda_{\text{IR}}}^{\lambda_{\text{UV}}} \frac{d\lambda}{\beta_\lambda}\right) \approx const.\exp\left(\frac{\pi^2}{\epsilon}\right), \tag{4.15}$$

where we integrated from $\lambda_{\text{UV}} = -O(1)$ to $\lambda_{\text{IR}} = +O(1)$. This agrees with the leading behavior of the exact result for $\xi_{\text{Potts}}$, found in [28] via the Bethe ansatz. As discussed in [1], at this order this prediction does not actually depend on knowing the OPE coefficient $C_{\varepsilon'\varepsilon'\varepsilon'}$, which cancels out of the answer, but only on $\varkappa$.[31] However, the ratios of $C_{\varepsilon'\varepsilon'\varepsilon'}$ to other OPE coefficients is significant as we will now see. Curiously, with an appropriate choice of an expansion parameter, the one-loop answer for $\xi_{\text{Potts}}$ turns out to be exact to all orders in perturbation theory, see the discussion around Eq. (4.45).

## 4.2 Im-flip for other operators

We will now perform the Im-flip check for other operators. At one loop, the anomalous dimension of a generic operator $\phi$ is given by

$$\gamma_\phi(g) = 2\pi C_{\phi\phi\varepsilon'}g, \tag{4.16}$$

---

[29]In [1] we used $\varkappa = -i\epsilon$, but here $\epsilon$ stands for $\sqrt{Q-4}$.

[30]The expression we used is given in the Appendix A of [35]. There this particular OPE coefficient is expressed as a finite product and hence as an analytic function of $m$. ($\rho$ of [35] is related to $m$ as $\rho = m/(m+1)$.).

[31]This can be seen already from the fact that $C_{\varepsilon'\varepsilon'\varepsilon'}$ can be scaled out of the beta-function (4.10) by rescaling $g$.

and the Im-flip condition reads [1]:

$$\frac{\text{Im}\Delta_\phi^{\mathcal{C}}}{C_{\phi\phi\varepsilon'}} = \frac{\text{Im}\Delta_{\varepsilon'}^{\mathcal{C}}}{C_{\varepsilon'\varepsilon'\varepsilon'}} = -\frac{\sqrt{3}}{2\pi}. \tag{4.17}$$

We will consider the following operators

- The energy operator $\varepsilon$:

$$\Delta_\varepsilon^{\mathcal{C}} = \frac{1}{2} - \frac{3i\epsilon}{4\pi} - \frac{3\epsilon^2}{8\pi^2} + \dots, \tag{4.18}$$

$$C_{\varepsilon\varepsilon'} = \frac{\sqrt{3}}{2} - \frac{i\sqrt{3}\epsilon}{4\pi} + \dots \tag{4.19}$$

- The spin operator $\sigma$:

$$\Delta_\sigma^{\mathcal{C}} = \frac{1}{8} - \frac{i\epsilon}{16\pi} + \dots, \tag{4.20}$$

$$C_{\sigma\sigma\varepsilon'} = \frac{1}{8\sqrt{3}} + \dots \tag{4.21}$$

Apart from orbifold at $Q = 4$, this OPE coefficient can be computed exactly for any $Q$ identifying $\sigma = \mathcal{O}_{\pm 1,0} = \phi_{\frac{m}{2},\frac{m}{2}}$ on the tricritical branch and continuing from the minimal models with $m$ even. Analytic continuation is straightforward because the Dotsenko-Fateev expression for $C(\phi_{\frac{m}{2},\frac{m}{2}}, \phi_{\frac{m}{2},\frac{m}{2}}, \phi_{1,3})$ contains a finite $m$-independent number of terms. This gives an OPE coefficient whose $m \to \infty$ limit agrees with the orbifold result. We won't need the subleading in $\epsilon$ terms so we don't quote them.

- The operator $Z \equiv \mathcal{O}_{0,1}$, contained in the Potts model with multiplicity $\frac{Q(Q-3)}{2}$, see section 2.6:

$$\Delta_Z^{\mathcal{C}} = 2 + \frac{i\epsilon}{\pi} + \dots, \tag{4.22}$$

$$C_{ZZ\varepsilon'} = -\frac{2}{\sqrt{3}} + \dots, \tag{4.23}$$

which does not appear in any of the minimal models,[32] but we can determine the $Q = 4$ OPE coefficient via the orbifold. Notice that the imaginary parts of $\Delta_Z^{\mathcal{C}}$ and of $C_{ZZ\varepsilon'}$ have the opposite sign compared to the all the other cases computed so far. This is related to the fact that $Z$ is irrelevant in the UV (on the tricritical branch for $Q < 4$), and relevant in the IR (i.e. on the critical branch), see Fig. 7.

As a sanity check, the Im-flip condition (4.17) is satisfied for all these operators by inspection. The success of this check can be traced to the fact that for $Q < 4$ we have the flow from tricritical to critical Potts models triggered by the same operator, $\phi_{1,3}$. Im-flip condition for $Q > 4$ is the analytically continued counterpart of the condition that ensures an appropriate change in the scaling dimensions of operators along this flow.

---

[32] For $Q$ such that $m$ is integer, this operator is actually Kac-degenerate with $r = m - 2$ and $s = m + 1$, however this is outside of the range of $s$ allowed for $\mathcal{M}_m$.

## 4.3 Drifting scaling dimensions

In this section we will use CPT to compute observables in the real physical theory in the range of distances corresponding to the walking regime, that is for $g = \frac{g_{FP}}{2} - \lambda$ with $\lambda$ small. We will discuss below the range of $\lambda$ for which our calculation is under control.

Consider the 2pt functions of some primary operator $\phi$. In the walking regime, we expect that the correlation functions exhibit approximate power-law scaling. To quantify this idea, we will define the drifting dimension of an operator $\phi$ as

$$\delta_\phi(r) = -\frac{1}{2} \frac{1}{G_\phi(r)} \frac{\partial G_\phi(r)}{\partial \log r}, \tag{4.24}$$

where $G_\phi$ is the 2pt function $\langle \phi(r)\phi(0) \rangle$. For a conformal theory, $\delta_\phi(r)$ would be just a constant equal to the scaling dimension, but in the walking regime it will be scale-dependent. In principle, it should be possible to measure the drifting dimensions, or at least closely related quantities, on the lattice as we mention below.

We compute $\delta_\phi(r)$ via the Callan-Symanzik (CS) equation. Restoring the dependence of the 2pt function on the renormalization scale $\mu$ and the renormalized coupling $g$, the CS equation for $G_\phi(r, g, \mu)$ reads:

$$\left[ \mu \partial_\mu + \beta(g)\partial_g + 2\gamma_\phi(g) \right] G_\phi(r, g, \mu) = 0. \tag{4.25}$$

To proceed we introduce the dimensionless variable $s = \mu r$ and factor out the fixed-point scaling of $G_\phi$:

$$G_\phi(r, g, \mu) = \frac{c(s, g)}{r^{2\Delta_\phi^C}}. \tag{4.26}$$

The solution of the CS equation for $c$ takes the well-known form

$$c(s, g) = \hat{c}(\bar{g}(s, g)) \exp\left[ -2 \int_1^s d\log s' \, \gamma_\phi(\bar{g}(s', g)) \right], \tag{4.27}$$

where $\bar{g}(s, g)$ is the running coupling with the initial conditions $\bar{g} = g$ at $s = 1$. We focus on the real RG trajectory $\bar{g} = g_{FP}/2 - \bar{\lambda}$ with initial condition $\bar{\lambda} = 0$ for definiteness. The running coupling satisfies $s\partial_s \bar{g} = -\beta(\bar{g})$, or equivalently $s\partial_s \bar{\lambda} = -\beta_\lambda(\bar{\lambda})$ with $\beta_\lambda$ given in (4.14). Integrating this equation we find

$$\bar{\lambda} = \frac{\sqrt{3}\epsilon}{4\pi^2} \tan\left( \frac{\epsilon \log s}{\pi} \right). \tag{4.28}$$

The one-loop contribution to $\hat{c}$, comes from the integrated 3pt function $\int d^2x \langle \phi \, \phi \, \epsilon'(x) \rangle$. After subtracting the $1/\epsilon$ pole, the $O(1)$ part of this integral vanishes,[33] so that

$$\hat{c} = 1 + O(i\epsilon)\bar{\lambda} + \dots. \tag{4.29}$$

At the one-loop order we can ignore the correction and set $\hat{c} = 1$. We will comment on this purely imaginary $O(\bar{\lambda})$ correction in section 4.4.

---

[33]The relevant integral takes the form $\int d^2x/(|x|^{2+\varkappa}|x - y|^{2+\varkappa}) = w(\varkappa)^2/w(2\varkappa)|y|^{2+2\varkappa}$ where $w(\varkappa)$ is the factor which arises when going to the Fourier transform $1/|x|^{2+\varkappa} \to w(\varkappa)|p|^{\varkappa}$. In the small $\varkappa$ limit $w(\varkappa) \sim 1/\varkappa + O(1)$ but it's easy to see that the $O(1)$ term will always disappear from the specific combination $w(\varkappa)^2/w(2\varkappa)$.

Using (4.16) we have

$$\gamma_\phi \left( \frac{g_{FP}}{2} - \lambda \right) = \pi C_{\phi\phi\varepsilon'} g_{FP} - 2\pi C_{\phi\phi\varepsilon'} \lambda. \tag{4.30}$$

It is easy to see that due to (4.17), the constant part of the anomalous dimension cancels exactly the imaginary part of $\Delta_\phi^{\mathcal{C}}$ in $1/r^{2\Delta_\phi^{\mathcal{C}}}$. Consequently, modulo an overall $r$-independent constant, $G_\phi$ will be real at this order, see below. Substituting (4.28) into (4.30) and doing the simple integral in (4.27) we finally get (denoting $\mu = 1/r_0$ where convenient)

$$G_\phi(r) = C(\mu) \frac{\left( \cos \frac{\epsilon \log r/r_0}{\pi} \right)^{-\sqrt{3} C_{\phi\phi\varepsilon'}}}{r^{2\,\mathrm{Re}\,\Delta_\phi^{\mathcal{C}}}}, \tag{4.31}$$

where $C(\mu) = \mu^{2i\,\mathrm{Im}\,\Delta_\phi^{\mathcal{C}}}$. While this factor is in general complex, we can absorb it defining the rescaled real operator $\phi_R$ by

$$\phi_R = \mu^{-i\mathrm{Im}\,\Delta_\phi^{\mathcal{C}}} \phi \,. \tag{4.32}$$

The two point function of $\phi_R$ is then real in the real theory. We conjecture that the same rescaling renders all higher $n$-point functions real as well, but at the moment this remains one of the future checks of our proposal. Real operators that have real correlation functions are thus related to operators naturally used in CPT by complex normalization factors which depend on the renormalization scale. We should not be surprised that this redefinition is needed to identify real operators, as it corresponds to an ambiguity in the choice of basis of the operators present in the complex theory $\mathcal{C}$.

Using (4.31), we compute the drifting dimension:

$$\delta_\phi(r) = \mathrm{Re}\,\Delta_\phi^{\mathcal{C}} - \frac{\sqrt{3}\epsilon}{2\pi} C_{\phi\phi\varepsilon'} \tan\left( \frac{\epsilon \ln r/r_0}{\pi} \right), \tag{4.33}$$

expressed in terms of the leading in $\epsilon$ values of the OPE coefficient and the operator dimension. For several operators $\phi$ of interest this complex CFT data was given in section 4.2.

Let us now discuss the range of validity of Eq. (4.33). First of all, there will be correction to (4.33) coming from ignoring the higher-order terms in the beta-function and anomalous dimension. Since the leading-order result for the deviation $\delta_\phi(r) - \Delta_\phi^{\mathcal{C}}$ is of order of the running coupling $\bar{\lambda}(r)$, we see that we can trust it as long as this deviation remains $\ll 1$ (for example there is no constraint that the deviation should remain $O(\epsilon)$ as one might have naively expected). This condition allows the argument of tangent in (4.33) to become $O(1)$ as long as it does not get within $O(\epsilon)$ to $\pi/2$. Another set of corrections to (4.33) will arise from the expansion of the CFT data in $\epsilon$, which are non-zero even at $\lambda = 0$.

To show a concrete example, we plotted in Fig. 12 the drifting dimension of the energy operator $\delta_\varepsilon$, as a function of the normalized logarithmic scale $x = \epsilon \log(r/r_0)$ for a few values of $Q$. We also indicate a (very rough) estimate of the theoretical error on this quantity. To estimate the $\lambda$-independent correction $[\delta_\varepsilon]_\epsilon$ we used the order $\epsilon^2$ contribution to $\Delta_\varepsilon^{\mathcal{C}}$ given in (4.18). On the other hand, the higher-loop correction was estimated as a relative correction to the non-trivial part of the drifting dimension proportional to the ratio of two-loop and one-loop terms in the beta-function (4.39) below, that is $[\delta_\varepsilon]_\lambda = (\delta_\varepsilon(r) - \mathrm{Re}\,\Delta_\varepsilon^{\mathcal{C}}) \frac{\beta^{(2)}}{\beta^{(1)}}$. We then add the corrections as mean squares $[\delta_\varepsilon]_{total} = \sqrt{[\delta_\varepsilon]_\epsilon^2 + [\delta_\varepsilon]_\lambda^2}$. Of course this is not meant to be a rigorous procedure, but just an estimate of the magnitude and qualitative behavior of the error.

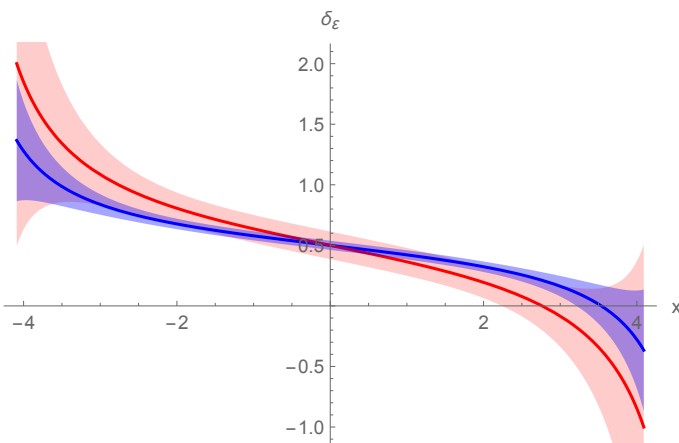

Figure 12: Drifting scaling dimension $\delta_\varepsilon$ given by the equation (4.33) as a function of $x = \epsilon \log(r/r_0)$ and an estimate of the theoretical uncertainty for $Q = 5$ (blue) and 7 (red).

Analogously to the drifting scaling dimension $\delta_\varepsilon(r)$ one can define a drifting exponent $\nu(r)$ as

$$\nu(r) = \frac{1}{d - \delta_\varepsilon(r)} \ . \qquad (4.34)$$

Recently a quantity similar to $\nu(r)$ was measured by means of lattice Monte Carlo simulations [43]. Some features of their Fig. 3 suggest qualitative agreement with (4.33), however, there is also some discrepancy. There is no immediate problem because the quantity they measured was not exactly (4.34), and different reasonable definitions of drifting exponents may not agree with each other. Notice in this regard that detailed behavior of drifting exponents is distinct even for two different definitions used in [43]. In the future it would be interesting to measure a quantity more directly related to (4.33) on the lattice. We believe that it should also be possible to perform an analytic calculation of finite-volume observables, such as those measured in [43], with the help of CPT around complex Potts model, but we leave this computation for the future.

The dependence of drifting dimensions on $\log r$ shown in Fig. 12, with an inflexion point at $r = r_0$, is pretty peculiar. It should be possible to use this dependence to differentiate the walking scenario from a more conventional scenario of nearly-scale invariant RG flow, namely the flow which slowly approaches an IR fixed point along a weakly irrelevant direction, with the schematic beta-function

$$\beta \sim \epsilon \lambda + \lambda^2 \ . \qquad (4.35)$$

For this flow, deviations of drifting dimensions from IR CFT limits will go like $\sim 1/\log(r)$ in the region $\epsilon \lesssim \lambda \ll 1$, smothly transitioning to a $const./r^\epsilon$ behavior at distances where $\lambda \lesssim \epsilon$. This functional dependence is clearly distinct from (4.33), in particular there is no inflection, and with enough precision it should be possible to distinguish the two scenarios.

Finally let us mention that drifting exponents do not seem related by simple analytic continuation to any $Q < 4$ quantity. To compute them it was important to first analytically continue the conformal data and then develop the CPT around the complex fixed point. We also stress that even if the exact values of CFT dimensions or OPE coefficients are not known, as for example is the case in all higher-dimensional examples of walking discussed in [1], the characteristic form of the drifting dimensions given by Eq. (4.33) stays the same and as such can be considered the smoking gun of the walking behavior.

### 4.4 Two-loop beta-function

Now we would like to go one order higher in $\epsilon$ and do perturbation theory up to two loops. We therefore need to address the question of scheme dependence. Up to two loops, we have a beta-function of the form

$$\beta^{\text{2-loop}} = \beta_1 g + \beta_2 g^2 + \beta_3 g^3. \tag{4.36}$$

Different schemes correspond to changes $g \to g + \alpha g^2 + \ldots$. As is well known, if $\beta_1 = 0$, the two-loop beta-function is scheme independent. However, in our case $\beta_1 \neq 0$, so we need to specify the scheme.

We will use the 'OPE scheme' [31], also used in [35].[34] The two-loop beta-function in this scheme takes the following form:

$$\beta^{\text{2-loop}} = \varkappa(\epsilon) g + \pi C_{\varepsilon'\varepsilon'\varepsilon'}(\epsilon) g^2 - \frac{\pi}{3} I_{\varepsilon'} g^3, \tag{4.37}$$

where as indicated we include the $\epsilon$-dependence of $\varkappa$ and of the OPE coefficient $C_{\varepsilon'\varepsilon'\varepsilon'}$.[35] Taking into account that $g_{FP} = O(\epsilon)$, at the two-loop order we should keep terms of total degree up to three in $\epsilon$ and $g$, which means that we will need $\varkappa$ at $O(\epsilon^2)$ and $C_{\varepsilon'\varepsilon'\varepsilon'}$ at $O(\epsilon)$, see (4.8),(4.11).

The two-loop coefficient $I_{\varepsilon'}$, needed at $O(1)$, originates from the "triple" OPE of the field $\varepsilon'$, i.e. divergences arising when three insertions of the field $\varepsilon'$ are close together. At this order it can be extracted from the integrated 4pt function $\int d^2 z \langle \varepsilon'(0) \varepsilon'(z) \varepsilon'(1) \varepsilon'(\infty) \rangle$ of the $Q = 4$ Potts model. However, one has to be careful and subtract extra divergences, which are related to the ordinary OPE of the field $\varepsilon'$ with itself or with other relevant operators. Taking into account all the subtractions, we have $I_{\varepsilon'} = -8\pi$.[36]

Computing the zero of the two-loop beta-function corresponding to $\overline{\mathcal{C}}$, it turns out that it's still given by (4.12), with no corrections at $O(\epsilon^2)$. This is a nice feature of the OPE scheme: were we to change $g \to g + \alpha g^2$ then $\text{Re} \, g_{FP} \neq 0$, losing the intuitive picture of $\mathcal{C}$ and $\bar{\mathcal{C}}$ sitting symmetrically around the real RG flow as in Fig. 11. That $g_{FP}$ remains purely imaginary is one of the reasons we chose the 'OPE scheme', the other being that the real RG stays a straight line passing halfway between the two complex fixed points, Eq. (4.13), see below. These desirable features would be lost at two loops in most schemes, e.g. in the scheme of [30].

Next we compute the dimension of $\varepsilon'$ at the $\overline{\mathcal{C}}$ fixed point, which comes out complex conjugate of $\varepsilon'$ in $\mathcal{C}$, as it should,

$$2 + \beta'(g)|_{g=g_{FP}} = 2 + \frac{2i\epsilon}{\pi} - \frac{\epsilon^2}{\pi^2}. \tag{4.38}$$

We now study the real walking RG trajectory. We claim that in our scheme it still given by $g = \frac{g_{FP}}{2} - \lambda$. To check this, we express the beta-function in terms of $\lambda$ and see that it comes out real:

$$\beta_\lambda^{\text{2-loop}}(\lambda) = -\frac{\sqrt{3}\epsilon^2}{4\pi^3} - \frac{4\pi}{\sqrt{3}} \lambda^2 + \frac{\epsilon^2}{2\pi^2} \lambda + \frac{8\pi^2}{3} \lambda^3. \tag{4.39}$$

---

[34] Eq. (4.39) equals $(-2)$ times Eq. (2.19) of [35] due to different normalization of beta-function used in that work..

[35] This is different e.g. from the scheme followed by [30], where the OPE coefficient in (4.37) is evaluated at $\epsilon = 0$. The two schemes are related by a coupling redefinition $g \to g + \alpha g^2$, which, at this order, only shifts the $\epsilon g^2$ term of the beta function. At this order, it does not change the value of $I_{\varepsilon'}$.

[36] This result was first obtained in [44] and then in [35], and we checked it numerically applying methods of [30,45] to the 4pt function $\langle \varepsilon'\varepsilon'\varepsilon'\varepsilon' \rangle$ given in appendix C of [35]. Minor modifications are required compared to [45], as it was tailored to a flow driven by an operator with a vanishing 3pt function.

It can also be checked that the anomalous dimension of $\varepsilon'$ has a constant imaginary part, which cancels exactly the imaginary part of its scaling dimension at $\mathcal{C}$:

$$\Delta_{\varepsilon'}(\lambda) = 2 + \beta'(g)|_{g=\frac{g_{FP}}{2}-\lambda} = 2 + \frac{\epsilon^2}{2\pi^2} - \frac{8\pi\lambda}{\sqrt{3}} + 8\pi^2\lambda^2. \qquad (4.40)$$

So in the OPE scheme, the real walking theory still corresponds to the line $\mathrm{Im}\, g = 0$ in the space of complex couplings $g \in \mathbb{C}$. While the Im-flip of $\varepsilon'$ at $\bar{\mathcal{C}}$ and cancellation of its imaginary part of the dimension on the real walking theory were automatic at one-loop, at two loops the check of these conditions explicitly involved the values of the OPE coefficients and integrated 4pt function $I_{\varepsilon'}$.

We can also consider other operators. For a generic operator $\phi$, the two-loop anomalous dimension is

$$\gamma_{\phi}(g) = 2\pi C_{\phi\phi\varepsilon'}g - \pi I_{\phi}g^2. \qquad (4.41)$$

As a check, for $\phi = \varepsilon'$ this agrees with $\Delta_{\varepsilon'}(g) = 2 + \beta'(g)$. At the considered order we need to keep terms with total degree up to two in $g$ and $\epsilon$. The $I_{\phi}$ is extracted from $\int d^2z \langle \phi(0)\varepsilon'(1)\varepsilon'(z)\phi(\infty)\rangle$ in the same way as previously explained for $I_{\varepsilon'}$.

Unfortunately the only other operator for which we currently have access to the 4pt function $\langle \phi\, \varepsilon'\varepsilon'\phi \rangle$ is $\phi = \varepsilon$ [35]. $C_{\varepsilon\varepsilon\varepsilon'}$ is given in (4.19), and we have $I_{\varepsilon} = -\pi$. Again it is easy to check the Im-flip at $\bar{\mathcal{C}}$ up to $O(\epsilon^2)$, as well as reality of the dimension of $\varepsilon$ on the real axis:

$$\Delta_{\varepsilon}(g_{FP}) = \Delta_{\varepsilon}^{\mathcal{C}} + \gamma_{\varepsilon}(g_{FP}) = \frac{1}{2} + \frac{3i\epsilon}{4\pi} - \frac{3\epsilon^2}{8\pi^2} + \ldots = \Delta_{\varepsilon}^{\bar{\mathcal{C}}}, \qquad (4.42)$$

$$\Delta_{\varepsilon}\left(\frac{g_{FP}}{2} - \lambda\right) = \Delta_{\varepsilon}^{\mathcal{C}} + \gamma_{\varepsilon}\left(\frac{g_{FP}}{2} - \lambda\right) = \frac{1}{2} - \frac{3\epsilon^2}{16\pi^2} - \sqrt{3}\pi\lambda + \pi^2\lambda^2 + \ldots \qquad (4.43)$$

Although we won't do this, we could use the given two-loop beta-function and the functions $\Delta_{\phi}\left(\frac{g_{FP}}{2} - \lambda\right)$ to extend the drifting dimension analysis of section 4.3 to the two-loop order. The provided information is sufficient to evaluate the integral $\int_1^s$ in (4.27), giving correction to drifting dimensions of relative order $O(1)\bar{\lambda}$. Since we verified that the functions entering the integral are real, the correction will be real, as expected in a real walking theory. The correction from $O(i\epsilon)\bar{\lambda}$ term in $\hat{c}$ in (4.29) is subleading. Since it is imaginary, we expect it to cancel with the imaginary part of the two-loop term in $\hat{c}$, although we have not checked.

## 4.5 General arguments about the real flow

We saw in section 4.1 that to the one-loop order in CPT reality of the theory at $g = g_{FP}/2 - \lambda$ is almost automatic. Let us now present some arguments why we expect to find a real theory to all orders in perturbation theory. In [1] we defined the complex conjugation map that acts on the space of QFTs, or equivalently on the space of RG flows. A real theory is then a fixed point of this map. At one loop we had the line of purely imaginary $g$ that connected $\mathcal{C}$ and $\bar{\mathcal{C}}$ and that was mapped into itself by the conjugation. Obviously any continuous map of an interval into itself has a fixed point. In general we expect that higher order corrections will deform the conjugation map in some continuous fashion, but under such deformations the fixed point is expected to remain. In general a continuous involution[37] possesses fixed points as long as the topology of the space on which it acts is relatively simple, see e.g. [46] and references therein for the precise theorem formulations. Since the topology of the space of theories in the vicinity

---

[37]Involution is a map which square is the identity, a condition that complex conjugation clearly satisfies.

of $\mathcal{C}$ and $\overline{\mathcal{C}}$ is trivial to leading order in CPT, it is likely to stay such under small deformations. Consequently the real theory should continue to exist in higher orders of perturbation theory.

In spite of this general argument, the above calculations showed that at the technical level emergence of the real observables is quite non-trivial. In section 4.4 we saw that quantities that do not directly correspond to physical observables, like anomalous dimensions and beta-functions, may actually be complex, and that this property depends on the choice of scheme. Calculation of section 4.3 demonstrated that, even at the one-loop order, one is required to use certain complex normalization constants in the definition of real operators. Nevertheless, both calculations do support the statement that the real theory exists and can be accessed by the deformation of the complex CFT.

## 4.6 The range of $Q$ for which the walking behavior persists

Having studied some higher-order results we can now try to gain some more intuition on the values of the parameter $Q$ for which we expect to see the walking behavior and consequently large correlation length in the Potts model. A rigorous discussion is possible in the limit $Q \to 4$, when the imaginary parts of operator dimensions go to zero, but this is not the only physically interesting regime, and moreover large values of the correlation length suggest that the walking regime extends all the way to $Q \sim 10$, see Table 1 in [1]. As it often happens, the $Q - 4$ expansion can be extended to large values of $Q$ after various factors of $\pi$ are taken into account. First of all, we see factors $\pi$ and $\pi^2$ in the one- and two-loop terms of the beta-function (4.37) or (4.39). These factors have purely geometric origin — they arise from angular integration, and the $n$-loop term will similarly carry factor $\pi^n$. This shows that a natural coupling constant along the real RG flow is $\tilde{\lambda} = \pi \lambda$.[38] In terms of this variable the beta-function reads

$$\beta_{\tilde{\lambda}} = -\frac{\sqrt{3}\epsilon^2}{4\pi^2} - \frac{4}{\sqrt{3}}\tilde{\lambda}^2 + \frac{\epsilon^2}{2\pi^2}\tilde{\lambda} + \frac{8}{3}\tilde{\lambda}^3. \tag{4.44}$$

We see that $\tilde{\lambda}^2$ and $\tilde{\lambda}^3$ coefficients became $O(1)$ numbers and we expect this to persist to higher orders in this normalization. We can also observe that the coefficients of different powers of $\tilde{\lambda}$ admit an expansion in $\epsilon^2/\pi^2$. It is the same expansion that we have seen above for complex CFT data, except that along the real flow only even powers of $i\epsilon/\pi$ enter. The fact that physical observables are expandable in even powers of $\epsilon$ follows from the fact that they are always real quantities, while the factors of $1/\pi$ can be traced back to (3.1) and should not be confused with the geometric $\pi$ rescaling performed above.

Consequently we expect the beta-function to maintain approximately the same form while the expansion parameter remains small: $\epsilon^2/\pi^2 \ll 1$. This identification of the expansion parameter makes it not so surprising that the picture obtained in small $\epsilon$ expansion remains reliable even for $\epsilon \sim \sqrt{6}$ corresponding to $Q \sim 10$. More generally, for perturbation around any complex CFT, the expansion will be controlled by the square of the imaginary part of the CFT dimension of the perturbing operator. In case at hand $(\mathrm{Im}\,\Delta_{\epsilon'})^2 \sim 4\epsilon^2/\pi^2$, and it appears that an extra factor of 4 does not affect the range of the validity of perturbation theory.

If we study the dependence of the complex CFT data on $\epsilon$ to higher orders than above, we realize that an even more natural expansion parameter is $4/|m|^2$, which agrees with $\epsilon^2/\pi^2$ to the first nontrivial order, see Fig. 13. Here $m = m(Q)$ is the 'minimal model' numbering parameter whose relation to $Q$ is given by (4.1), so that $g(Q) = 4 + 4/m(Q)$, $m(Q)$ is real for $Q \leqslant 4$ and becomes purely imaginary for $Q > 4$.

---

[38]This is similar to counting $1/(16\pi^2)$ factors appearing in Feynman-diagrammatic perturbation theory in 4d.

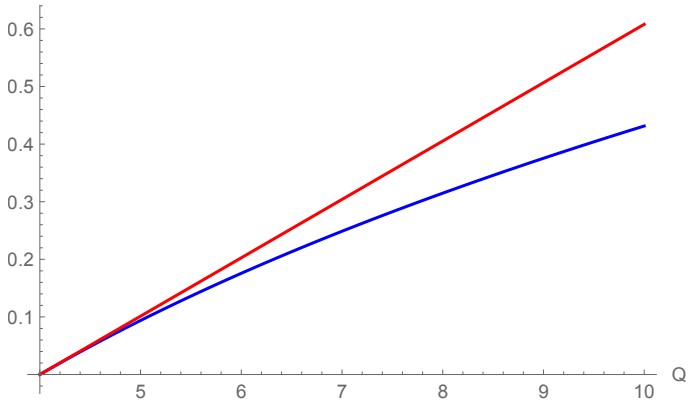

Figure 13: Expansion parameters $\frac{\epsilon^2}{\pi^2}$ (red) and $\frac{4}{|m|^2}$ (blue).

At this point it would be nice to compute some physical observable along the walking flow, like a drifting dimension, to higher order in $\epsilon/\pi$ or $1/m$ and to confirm the suggested behavior. Unfortunately, such a computation appears to be rather tedious since the first nontrivial correction is expected at relative order $1/m^2$, requiring to compute at NNLO (three loops). Alternatively, we could study the behavior of perturbation theory by expanding in the coupling constant some exact non-perturbative result. One such quantity is the correlation length on the square lattice computed exactly for any $Q$ in [28]. Computation of this quantity in perturbation theory using Eq. (4.15) would include two types of corrections, those coming from the beta-function and those coming from the dependence of $\lambda_{\text{UV}}$ and $\lambda_{\text{IR}}$ on $Q$. Inspection of the exact answer, however, leads to a big surprise: if $1/m$ is used as an expansion parameter the answer turns out to be one-loop exact! Namely, expanded at small $1/|m|$ the exact correlation length reads[39]

$$\xi_{\text{Potts}} = \frac{\sqrt{2}}{16} e^{\pi|m|/2} + O\left(e^{-\pi|m|/2}\right). \tag{4.45}$$

We see that all corrections to the one-loop answer are non-perturbative in $1/m$. As far as we know this observation has not been made before. This fact is absolutely non-manifest in our perturbation theory, and it suggests that there might be an even superior computational scheme. We leave exploration of this possibility for the future.

## 5    Conclusions

In this paper we carried on with our proposal [1] that walking behavior can be understood as an RG flow passing between two complex CFTs. Here we were able to provide an example where we have access to the complex CFTs, meaning that we know the operator spectrum and many of the OPE coefficients. The model we studied is the 2d Potts model, which undergoes a weakly first-order phase transition and walking behavior when the number of states $Q$ is in the range $4 \lesssim Q \lesssim 10$. For $Q < 4$, on the other hand, the model has a critical and a tricritical point, and the corresponding partition functions on the torus, and hence the operator spectrum, are known [3]. We access the full complex CFT spectrum by analytically continuing these partition functions to $Q > 4$. We are also able to analytically continue some OPE coefficients involving Kac-degenerate operators, and determine a few others near $Q = 4$ via the orbifold construction of the $Q = 4$ Potts model.

---

[39]See Eq. (4.46) of [28]; there is an obvious typo in (4.47).

We can then describe the real walking theory as the complex CFT perturbed by a nearly marginal operator, making predictions for observable quantities. Since the walking regime is only approximately scale invariant, 2pt functions exhibit small deviations from power laws, which we compute using perturbation theory. It would be interesting to check our results for drifting scaling dimensions with lattice measurements. These techniques also allow to compute the correlation length of the model for $Q \gtrsim 4$, in agreement with the result obtained using integrability.

The construction passes several purely theoretical consistency checks as well. It was anticipated in [1] that the existence of two complex conjugate CFTs next to each other, as well as of the real theory in between them, requires certain conspiracies in the conformal data. Here we confirmed, up to the two-loop order, that these conspiracies indeed take place in the complex Potts CFTs.

We would like to emphasize that our complex CFTs are not just some approximations which may break down upon closer look. On the contrary, they are perfectly nonperturbatively well defined theories which satisfy usual CFT axioms (OPE, crossing, modular-invariant partition function). Being non-unitary, they are naturally defined in Euclidean signature, and may not necessarily allow analytic continuation to Lorentzian signature.[40] We outlined how to look for them by studying the Potts model in the space of complexified couplings.

There is at least one more example of walking in two dimensions for which some exact information can be extracted about the complex CFTs, thus providing further tests and applications of our idea: the $O(n)$ model. Very similarly to the Potts model, for $-2 \leqslant n \leqslant 2$ there are two branches of fixed points, the critical and the low-temperature fixed point (which is in general non-trivial). At $n = n_c = 2$ the two fixed points merge and go into the complex plane. At $n > 2$ there is no phase transition, nevertheless, for $n \gtrsim 2$ we expect to find a massive theory without any tunable parameter with large correlation length due to walking RG behavior. Many of our techniques can be applied to the $O(n)$ model in an analogous way. This theory, especially for $n$ integer, has been extensively studied in the past, however, to the best of our knowledge, the walking behavior has not been emphasized (see however [47]). Notice that the walking behavior for $n \gtrsim 2$ would be a distinct phenomenon from the slow logarithmic running of the nonlinear sigma-model coupling at short distances. It remains to be seen how far above $n_c = 2$ the walking regime extends, and whether any vestiges of walking remain visible at $n = 3$.

Walking can also be realized in higher dimensions, and in [1] we presented several examples: 3d and 4d gauge theories below conformal window, the three-state Potts model in 3d, and possibly deconfined criticality transitions. Certainly there are others. In these theories it is harder to make quantitative predictions, since the analytic computations of the properties of corresponding complex CFTs are less feasible. Still, the mechanism governing the RG behavior is the same. Our ability to derive the properties of the walking flow from given conformal data is independent of the number of dimensions. Hence several our results, like the form of the walking 2pt function, can be immediately transcribed for the higher-dimensional cases.

---

[40]The Osterwalder-Shrader theorem allowing analytic continuation from Euclidean to Lorentzian only works for reflection-positive, i.e. unitary, theories.

## Acknowledgements

We thank Damon Binder, Dmitry Chelkak, Jesper Jacobsen and Hubert Saleur for very useful discussions, and John Cardy, Bernard Nienhuis, and Jean-Bernard Zuber for comments on the draft. VG is grateful to CERN and ENS for hospitality. We are all grateful to Caltech for hospitality during the completion of this work, and to the organizers and participants of "Bootstrap 2018" for their interest and comments. We are grateful to Sylvain Ribault for careful refereeing of our paper for SciPost, and many comments which led to an improvement of presentation.

SR is supported by the Simons Foundation grant 488655 (Simons Collaboration on the Nonperturbative Bootstrap), and by Mitsubishi Heavy Industries as an ENS-MHI Chair holder. BZ is supported by the National Centre of Competence in Research SwissMAP funded by the Swiss National Science Foundation.

## A  Kondev's argument

Let us discuss the calculation of Kondev [23], whose purpose was to determine the dependence of the coupling $g$ on $Q$ given in (2.7) without relying on non-trivial exact solvability results about the $F$-model.

So let us pretend that we don't know (2.7) and follow the discussion from Eq. (2.5) to (2.16). The discussion still works, except we should not use (2.12) which was based on (2.7). Then Kondev demands that the operator $e^{-i4\phi}$ whose dimension is given by (2.16) for $e = -4$ should be 'exactly marginal'. Ref. [23] gives some physical arguments in favor of this claim, having to do with loops of various sizes having the same weight. We do not fully understand these arguments, and as discussed below we have reasons to doubt that this interpretation is correct. Still, if we accept this observation, then from $\Delta(e^{-i4\phi}) = 2$ we obtain (2.12) as a consequence, and from (2.12) and (2.5) we obtain (2.7).[41]

While Kondev's calculation has definitely some truth to it, and is cited in many works, we believe the correct interpretation is different. First of all, the spectrum of minimal models describing the Potts model for $Q = 2, 3$ certainly does not contain any marginal operators. Moreover, as explained in section 2.6, the operator $e^{-i4\phi}$ is not actually present in the torus partition for any $Q$. We have checked that it is also absent in the cylinder partition function as computed in [19, 20] and expanded in the channel along the cylinder axis in [20]. So for all we know this operator does not exist in the theory.

Another reason for doubt is as follows. Kondev mentions that the operator which he wants to be 'exactly marginal' is responsible for the loop weight, and requires that it should not renormalize, and hence be marginal. Following this logic, it appears that by changing the coefficient of this operator in the action we could modify $Q \to Q + \delta Q$. As is well known, two CFTs related by perturbing with an exactly marginal local deformation have the same central charge. Since the Potts model central charge depends on $Q$, the above construction should be impossible, which confirms that the $e^{-i4\phi}$ operator should not exist for any $Q$.

One could try to resolve these puzzles by saying that Kondev's operator is in fact nonlocal. More concretely, one could identify it with the nonlocal marginal "operator" conjugate to $Q$

---

[41]Sometimes Kondev's argument is stated as saying that one of the two operators $e^{\pm i4\phi}$ should be marginal, with two choices leading to equivalent solutions with $e_0 \to -e_0$.

in the continuous family of CFTs.[42] Because of nonlocality, central charge could vary when perturbing by this operator, nor would it have to appear in the partition function. We do not however believe that Kondev's argument leaves room for such an identification. Indeed, Kondev [23] and related works treat operator $e^{-i4\phi}$ as just another electric vertex operator whose dimension is given by a Coulomb gas formula. The other operators of such type, when they exist, are perfectly local, so by extension $e^{-i4\phi}$ should also be local. If $e^{-i4\phi}$ were nonlocal, then this would raise the puzzle why the other electric operators are local.

We would like to save the computation behind Kondev's argument, by giving it the following new interpretation. The Di Francesco-Saleur-Zuber (DFSZ) computation leading to the Potts model partition function (2.26) did not depend on (2.7). Suppose we don't assume (2.7), can we derive it from (2.26)? This is indeed possible. As we saw in section 2.6, consistency of Eq. (2.26) with the form of the partition function expected in a CFT is not automatic. Namely, the character of the unit operator appears to have a vector state at level 1 and a scalar state at level 2, while the true character should not contain any states of this type. The removal of these unwanted states requires the presence, among the rest of the Coulomb gas dimensions and spins, of a spin-1 operator of dimension 1 and multiplicity −1, and a scalar operator of dimension 2 and multiplicity 1. The latter scalar 'operator' is precisely the one appearing in the original Kondev's argument.

The bottom line is that the $g$-$Q$ relation can be fixed based on the following assumptions: Coulomb gas dimension, DFSZ partition function, and the absence of descendants of the unit operator on levels 1,2 ($\partial_z \mathbb{1} = \partial_{\bar{z}} \mathbb{1} = \partial_z \partial_{\bar{z}} \mathbb{1} = 0$). The algebraic computation is the same as Kondev's, but the interpretation is the opposite. Instead of requiring that an 'exactly marginal' operator exist in the spectrum, one requires that there should be no such operator once all pieces of the partition function are taken into account.

## B  Representations of $S_Q$ and the operator spectrum

Irreps of $S_Q$ are in one-to-one correspondence with Young tableaux $Y$ with $Q$ boxes, and their dimension $D_Y(Q)$ is given by the hook rule [48]:

$$D_Y(Q) = \frac{Q!}{\prod\limits_{\text{boxes}} \text{hook length}} . \tag{B.1}$$

The hook length for one given box is the number of boxes below it, plus the number of boxes to its right, plus one. For example, the irrep

$$\underbrace{\begin{array}{|c|c|c|c|c|c|}\hline & & & & & \\\hline\end{array}}_{Q \text{ boxes}} \tag{B.2}$$

has dimension $\frac{Q!}{Q!} = 1$, and is the singlet representation of $S_Q$. The irrep

$$\underbrace{\begin{array}{|c|c|c|c|c|}\hline & & & & \\\hline & \\\cline{1-1}\end{array}}_{Q-1 \text{ boxes}} \tag{B.3}$$

has dimension $\frac{Q!}{Q(Q-2)!} = Q-1$ and is the vector representation.[43]

---

[42]We thank Bernard Nienhuis for a discussion.

[43]$S_Q$ acts naturaly on the $Q$-dimensional vector space with the basis $\phi_a$ ($a = 1, \ldots, Q$), by permuting the indices. This representation is reducible and decomposes into singlet $\sum_a \phi_a$, and $(Q-1)$-dimensional vector $\tilde{\phi}_a = \phi_a - \frac{1}{Q}\sum_b \phi_b$, $\sum_a \tilde{\phi}_a = 0$.

By the hook rule, $D_Y(Q)$ is a polynomial in $Q$ with integer single zeros. Now we list the dimension of a few irreps of $S_Q$. Denote by $[a_1, a_2, \ldots, a_n]$ the Young tableau with $a_1$ boxes in the first row, $a_2$ boxes in the second row, etc., so that $\sum_i a_i = Q$. Then we have ($(a)_n$ is the Pochhammer symbol)

$$D_{[Q]} = 1, \tag{B.4}$$

$$D_{[Q-n,n]} = \frac{(Q-n+2)_{n-1}}{n!}(Q-2n+1) \quad \text{with} \quad n \leqslant \left\lfloor \frac{Q}{2} \right\rfloor, \tag{B.5}$$

$$D_{[Q-n,1,\ldots,1]} = \frac{(Q-n)_n}{n!} \quad \text{with} \quad n < Q, \tag{B.6}$$

$$D_{[Q-4,2,1,1]} = \frac{Q(Q-2)(Q-3)(Q-5)}{8}. \tag{B.7}$$

We now test the observation from section 2.6: *Multiplicities $M_{h,\bar{h}}$ can be decomposed as sums of dimensions of irreps of $S_Q$, analytically continued in $Q$, with $Q$-independent positive integer coefficients.* This is nontrivial: $M_{h,\bar{h}}$ is a polynomial in $Q$, but not every polynomial has the stated property. The simplest counterexample would be $M_{h,\bar{h}} = Q - 2$. Notice that it would still be consistent with having true $S_Q$ symmetry for integer $Q \geqslant 2$, were we to relax the request for $Q$-independence of the expansion coefficients and let the number of singlets scale with $Q$ as $Q - 2$. The latter realization of the symmetry, however, appears less elegant.

A few operators whose multiplicities coincide with a dimension of a single irrep ($\mathbb{1}$, $\varepsilon$, $\varepsilon'$, $\sigma$, $\sigma'$, $\mathcal{O}_{0,1}$) were discussed in section 2.6. One more example is the spin-1 operator $\mathcal{O}_{1,1}$, whose multiplicity is

$$\Lambda(2,2) + Q - 1 = \frac{1}{2}(Q-1)(Q-2) = D_{[Q-2,1,1]} = D_{[Q-2,2]} + 1. \tag{B.8}$$

The latter identity means that $\mathcal{O}_{1,1}$ multiplet may also decompose into two irreps.

In more complicated case more than one irrep is required. E.g. consider the scalar $\mathcal{O}_{0,3/2}$, which comes from the term with $M = 3, N = 1, P = 0$ in the second term of (2.23), as well as from $Z_c[g, 1/2]$. This operator has multiplicity $\Lambda(3,1) - (Q-1) = \frac{(Q^2-5Q+3)(Q-1)}{3}$. This quantity is clearly not the dimension of an irrep of $S_Q$, since it has zeros at non-integer values of $Q$. However it can be decomposed as a sum of dimension of irreps of $S_Q$. We have two possibilities:

$$\frac{(Q^2-5Q+3)(Q-1)}{3} = D_{[Q-3,1,1,1]} + D_{[Q-3,3]} = D_{[Q-1,1]} + 2D_{[Q-3,3]}. \tag{B.9}$$

Another interesting operator is $\mathcal{O}_{0,2}$, which arises from the $M = 4, P = 0, N = 1$ term of (2.23). Its multiplicity is $\Lambda(4,1) = \frac{1}{4}Q(Q-2)(Q-3)^2$ which has a double zero and hence is not a dimension of an irrep of $S_Q$. Here's one of several possible decompositions as a sum of irreps:

$$\frac{Q(Q-2)(Q-3)^2}{4} = 3D_{[Q-3,1,1,1]} + D_{[Q-4,2,1,1]} + 3D_{[Q-4,1,1,1]}. \tag{B.10}$$

Whenever there are multiple possible decompositions, we cannot currently decide which one is physically preferred. Hopefully this can be done in the future by defining some sort of twisted partition function, which would give different weights to different irreps, or by studying the structure of the 3pt functions.

Let's finally mention what happens for $Q = 2, 3, 4$. As mentioned in section 2.6, all multiplicities for these $Q$ should be positive. The case of operator $\mathcal{O}_{0,1}$, which might appear to have

negative multiplicity for $Q = 2$, was already discussed in section 2.6. Similarly, operator $\mathcal{O}_{0,3/2}$ appears to have negative multiplicity for $Q = 2, 3, 4$, but this is resolved by degeneracies with other operators of the theory.[44] These are examples of a general phenomenon: for $Q = 2, 3, 4$ partition function (2.26) can be transformed to a simpler form [3], showing manifestly that the theory contains primaries with positive multiplicities only.

## C $\;Q = 4$ Potts model as an orbifold free boson

The $Q = 4$ Potts model can be described as free scalar boson compactified on $S_1/\mathbb{Z}_2$ with radius $R = 1/\sqrt{2}$, see e.g. [41] whose notation we will follow. Local operators in this theory are built from holomorphic and anti-holomorphic components of the scalar field $\phi$ and $\bar{\phi}$.[45]

First, let us identify the operator $\varepsilon'$ in this description. In total there are three marginal operators [41]:

$$L = \partial \phi \, \bar{\partial} \bar{\phi}, \quad V_1 = V_{0,4}^+ = \sqrt{2} \cos\left(\sqrt{2}(\phi - \bar{\phi})\right), \quad V_2 = V_{1,0}^+ = \sqrt{2} \cos\left(\sqrt{2}(\phi + \bar{\phi})\right). \quad \text{(C.1)}$$

Consequently $\varepsilon'$ must be a linear combination of these. To determine which one, let us use the $\varepsilon \times \varepsilon$ OPE. Since the remaining marginal operators transform non-trivially under $S_4$ and $\varepsilon$ is a singlet, the only dimension 2 operator that can appear in this OPE is $\varepsilon'$. The energy operator itself can be easily identified in the orbifold theory: $\varepsilon = V_{0,2}^+$, since this is the only operator of dimension 1/2.

OPEs of vertex operators and $L$ are well known (see e.g. [49]). In particular,

$$L \times V_{n,m}^+ \sim \left(\frac{n^2}{R^2} - \frac{m^2 R^2}{4}\right) V_{n,m}^+, \quad V_{n,m}^+ \times V_{n',m'}^+ \sim \frac{1}{\sqrt{2}} V_{n+n',m+m'}^+. \quad \text{(C.2)}$$

We get

$$V_{0,2}^+ \times V_{0,2}^+ \sim \frac{1}{\sqrt{2}} V_{0,4}^+ - \frac{1}{2} L = C_{\varepsilon\varepsilon'} \varepsilon', \quad \text{(C.3)}$$

where

$$\varepsilon' = \frac{\sqrt{2}}{\sqrt{3}} V_{0,4}^+ - \frac{1}{\sqrt{3}} L \quad \text{(C.4)}$$

is unit-normalized, and $C_{\varepsilon\varepsilon'} = \frac{\sqrt{3}}{2}$ agrees with (4.19). Using (C.4), we also confirm the $O(\epsilon^0)$ term in (4.11).

Combinations of $L$, $V_1$ and $V_2$ orthogonal to $\varepsilon'$ are the remaining marginal operators:

$$Z_1 = V_2, \quad Z_2 = \frac{2}{\sqrt{3}} L + \frac{1}{\sqrt{3}} V_1. \quad \text{(C.5)}$$

This leads to $\varepsilon' \times Z_i \sim -\frac{2}{\sqrt{3}} Z_i$, and hence to (4.23).

Turning to the spin operator, one of its components is identified with $V_{0,1}^+$ (the other two residing in the twisted sector). Using (C.2) we get

$$V_{0,1}^+ \times V_{0,1}^+ \sim -\frac{1}{8} L, \quad \text{(C.6)}$$

---

[44]On the critical branch it is degenerate with $\mathcal{O}_{e_0+4,0}$, $\mathcal{O}_{\pm5,0}$ and $\mathcal{O}_{e_0\pm6,0}$, for $Q = 2, 3, 4$ correspondingly. The same happens on the tricritical branch, but with different operators ($\mathcal{O}_{e_0+8,0}$, $\mathcal{O}_{\pm7,0}$ and $\mathcal{O}_{e_0\pm6,0}$).

[45]Notice that this is not the same scalar field as the height field denoted by the same letter in section 2.3. It is tempting to say that the two fields are related by some sort of T-duality, but the precise relation is unclear to us due to the necessity to orbifold. In a simpler case of the $O(2)$ model, the height field used in [3] is indeed the T-dual of the usual compactified boson.

which after taking the inner product with $\varepsilon'$ leads to (4.21).

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
