# Peer review of "Walking, Weak first-order transitions, and Complex CFTs II. Two-dimensional Potts model at $Q>4$"

_SciPost Physics, doi:SciPost Phys. 5, 050 (2018)_

## Round 2 · Referee Report · Sylvain Ribault (Referee 1) · 2018-9-17

Strengths
1- Predicting and checking the nontrivial Im-flip condition for CFT data.
2- Good review of the 2d Potts model.
3- A foray into the oft-neglected 2d CFTs with complex central charges.
Weaknesses
1- Clarity and precision are sometimes lacking.
Report
The main reason why I am confident that the article's picture of walking RG flows is correct, is that it provides predictions for relations between conformal dimensions and three-point structure constants: the Im-flip condition of Section 4.2. The authors check this condition in a few nontrivial cases, and I believe that it holds more generally, based on rough calculations.
Other reasons why the article is interesting are a good (but perfectible) review of the 2d Potts model, and the foray into the oft-neglected CFTs with complex central charges.
I believe that the text's clarity could and should be improved in many places. Although the general ideas are clear enough, the technical implementation is sometimes hard to follow. In my suggestions I will first focus on three areas that deserve particular attention, before giving miscellaneous comments on the rest of the text.
NOTE ADDED AFTER VETTING: I am grateful to the authors for their explanations, which greatly helped me write this report.
Requested changes
(For a better formatted and printable version, see the attached PDF.)
\subsubsection*{Sections 2.4, 2.5 and 2.6}
These Sections review the spectrum of the Potts model. While only some features of this spectrum will be needed later, this review is valuable in itself. I believe that its clarity and presentation should be improved along the following lines:
1- We have no less than 7 different notations for what is essentially the same parameter: $c, Q, e_0, g, u, t, m$. Using several notations is probably inevitable, as this parameter appears in various contexts and approaches. But 7 notations are too many, maybe $Q,c,g$ or $Q,c,t$ could suffice? If many notations are kept, at least there should be a synthetic table of their relations.
2- Similarly, we have two notations $(e,m)$ and $(r,s)$ for the same parameters. The discussion of Section 2.6 on how degenerate characters emerge from various terms in the partition function would be greatly helped by using $(r,s)$ throughout, as degenerate representations correspond to $r,s$ being nonzero integers of the same sign. Or at least, the partition function should be rewritten in terms of $(r,s)$ in Section 2.6.
3- The formulas (2.15) and (2.16) are standard formulas in free bosonic CFTs in the presence of a background charge. The present text gives the misleading impression that these formulas are specific to the Potts model, and gives derivations that are not very enlightening and probably unnecessary. Once we know that we should compute the functional integral (2.13), these formulas can be immediately accepted, with possibly a reference to the textbook by di Francesco et al, Chapter 9.
4- If $e\in 2\mathbb{Z}$ for electric operators (Section 2.4), why is $e\in \mathbb{Z}$ allowed in $Z_c[g,1]$ (2.23)-(2.19)?
5- It is strange that the partition function (2.23) is only partially described in Section 2.5, with one piece being postponed to eq. (2.31). Some restructuring would be welcome.
6- $1$ is not prime: is (2.32) valid for $M=1$?
7- The notation $\mathcal{O}_{e_0+2P,0}$ is confusing, and obscures the fact that this operator is degenerate if $P\geq 0$. A notation of the type $V_{1,P+1}$ would be much better. (I used the letter $V$ in order to avoid the confusion with the non-diagonal operators $\mathcal{O}_{e,m}$.) Similarly, in (2.31) the notation $x_{e_0+2P,0}$ is confusing.
8- When discussing the operator content, in other words when grouping the terms in the partition function into Virasoro characters, it would be good to systematically indicate which contributions come from which terms with which multiplicities. Presenting all the results as a table might be appropriate.
9- Statements like 'does not exist as a primary operator' or 'a scalar operator of such dimension is present, but it's not a primary' or 'do not exist in the spectrum' should be made more precise. In the presence of negative multiplicities, it is hard to say anything definite about the spectrum: having statements about characters would be more accurate. The appearance of degenerate characters suggests, but does not prove, that degenerate operators exist. And null vectors are both primary and descendent.
10- In Figure 7, at $Q=2$ there seem to be two degenerate operators of type $(1,2)$ with different dimensions.
11- The operators $\epsilon,\epsilon'$ are actually degenerate at all $Q$, whereas Figure 7 suggests that they are degenerate at $Q=2,3$ only. The indices $(r,s)$ are not just positions in the Kac table: they make sense at all central charges.
\subsubsection*{Section 4 before 4.1}
The authors could be more cautious in declaring that the light fields belong to degenerate conformal families: the characters suggest it, but do not prove it. Character-wise, a degenerate representation coincides with two non-degenerate representations with multiplicities $1$ and $-1$, and we are in a context where negative multiplicities are allowed.
Moreover, fixing three-point functions of degenerate fields at arbitrary central charges using crossing symmetry is not really complicated. The authors' extrapolation from minimal models is a circuitous route, which gives correct results because the correlators are uniquely determined by differential equations and crossing relations, as the authors correctly state. But this holds for all central charges.
Therefore, I recommend the following:
12- Remove the discussion of minimal models, with its odd focus on the unitary models with integer $m$, whereas more general minimal models have two integer parameters. (The mention of minimal models in the article's Abstract could also be removed.)
13- Give the explicit formulas for the structure constants, as this would allow readers to check their subsequent expansions around $c=1$. (See footnote 24.) The natural parameter for writing these structure constants is not $m$, but rather $t$ or $g$. Maybe these formulas could be given in an appendix, which would include all needed structure constants, including those of the spin operator.
\subsubsection*{Section 4.5}
This Section gives arguments for the existence of the real flow, based on the assumption that there exists a pair of complex conjugate CFTs that are moreover related by a RG flow. There are two arguments:
A first-order argument argument, plus a heuristic argument that higher order corrections should deform the picture without changing it qualitatively.
An all-orders argument, which is not rigorous because regularization issues are neglected, and which essentially shows the equivalence between the existence of the real flow, and the fixed point coupling constant $g_{FP}$ being pure imaginary.
The all-orders argument does not really establish the existence of the real flow, but it does constitute a nontrivial consistency check. Moreover, it illustrates the properties of correlation functions in complex CFTs.
The logic of the argument, and the technical details, are not clear enough. Moreover, while interesting, this Section is not crucial to the main results of Section 4. I recommend that Section 4.5 be either deleted, or clarified. Suggested clarifications include:
14- Removing the misleading sentence 'a formal perturbative argument for the existence of the real theory', and clarifying the sentence after (4.49) as establishing '$X\Leftrightarrow Y$' rather than '$X$ and $Y$'.
15- Explaining the precise meaning of the two relations between the CFTs $\mathcal{C}$ and $\bar{\mathcal{C}}$, at the level of correlation functions: the relation that they are complex conjugates, and the relation via the RG flow.
16- 'regulate' $\to$ 'regularize'.
17- Typo in (4.47): $g_2^k$. Same problem in (4.48).
18- Better explaining the crucial step of going from the first to the second line of (4.47).
19- Clarifying that at each point on the RG flow, there exists only one operator $\Phi$ or $\Phi^*$, which however changes name from $\Phi$ to $\Phi^*$ at some point, and which becomes real in a real CFT.
\subsubsection*{Miscellaneous suggestions}
20- Does eq. (2.4) for the partition function at arbitrary $Q$ reduces to (2.2) when $Q$ is integer?
21- In Subsection 2.2, clarify the statement that 'we will take an intuitive approach to symmetry for non-integer $Q$...'. Does this symmetry for non-integer $Q$ play any role in the subsequent analysis?
22- Typo: 'complex plain'.
23- In Section 3, the statement that 'multiplicities stay real for any $Q$' could be amended to 'multiplicities stay real for any real $Q$'. Thinking about complex $Q$ is indeed quite natural in this context.
24- There is a dual use of the notation $g$ in Section 4.1: the original $g$ from eq. (2.6), which appears again in eq. (4.7), differs from the $g$ in eq. (4.9).
25- In Section 4.1, the description of the Im-flip is confusing. Maybe an equation would be clearer.
26- In the Conclusion, state explicitly what it means to 'make sense' for a Euclidean CFT, and for a Minkowskian CFT.

---

## Round 3 · Referee Report · Sylvain Ribault · 2018-10-23

Report

The authors' use of my suggestions for improvement was thorough and economical. Thorough, because they considered each suggestion, and decided which modifications to perform (if any) on a case by case basis, while carefully reporting the changes in their reply. Economical, because each change tends to be the smallest modification that could do the requested clarification: in many cases this meant adding a footnote. The resulting changes are local and perturbative, except for the all-orders argument of Section 4.5, which the authors chose to delete rather than clarify.

The resulting text is still not an easy read, but the specific sources of confusion and misunderstanding that I had identified have been eliminated.

Let me briefly comment on two of the few suggestions that the authors chose not to follow:
- I now see that my remark 10 was misguided, as indeed the critical and tricritical models with the same $Q$ have different central charges.
- My suggestion 12 to eliminate minimal models can indeed be ignored, provided readers take the authors to the letter in considering the use of minimal models as a 'helpful trick' only (page 17).

---

## Round 3 · Referee Report · Anonymous · 2018-10-30

Report

The revised version answers all criticism put by the two referees. The paper can now be published.
A typo is spotted in reference [44] - the name of the auther is misspelled.

---

## Round 3 · Author Response

We thanks the referees for their comments. We especially thank the second referee Sylvain Ribault for his exceptionally thorough reading of our paper and for many comments.

---

## Round 3 · List of Changes

REPLIES TO SUGGESTIONS OF ANONYMOUS REFEREE:
I would like to bring to the attention of the authors two papers in which two loop integrals of conformal perturbation theory are calculated for nearby fixed points (they appear to be essentially the same as those in section 4.4):
• A. W. W. Ludwig and J. Cardy, Nuclear Physics B285, 687
• M. Lassig, Nuclear Physics B334, 652

OUR REPLY: Checked and one of the references added in note 36

A couple of typos were spotted:
• On page 15, next to last sentence in section 3: ”form” should be changed to ”from”

OUR REPLY: taken care

• On page 20, just before formula (4.22) an ”echo” of ”with” should be removed

OUR REPLY: taken care
* * *
REPLIES TO QUERIES/SUGGESTIONS OF SYLVAIN RIBAULT:
1- We have no less than 7 different notations for what is essentially the same parameter: c,Q,e0,g,u,t,m. Using several notations is probably inevitable, as this parameter appears in various contexts and approaches. But 7 notations are too many, maybe Q,c,g or Q,c,t could suffice? If many notations are kept, at least there should be a synthetic table of their relations.

OUR REPLY: Table 1 created
* * *
2- Similarly, we have two notations (e,m) and (r,s) for the same parameters. The discussion of Section 2.6 on how degenerate characters emerge from various terms in the partition function would be greatly helped by using (r,s) throughout, as degenerate representations correspond to r,s being nonzero integers of the same sign. Or at least, the partition function should be rewritten in terms of (r,s) in Section 2.6.

OUR REPLY: : We kept our notation. See new footnote 11. We do not rewrite the partition function fully in terms of h_{rs} characters in this paper, see the sentence added before “Singlets” subsection 2.6.
* * *
3- The formulas (2.15) and (2.16) are standard formulas in free bosonic CFTs in the presence of a background charge. The present text gives the misleading impression that these formulas are specific to the Potts model, and gives derivations that are not very enlightening and probably unnecessary. Once we know that we should compute the functional integral (2.13), these formulas can be immediately accepted, with possibly a reference to the textbook by di Francesco et al, Chapter 9.

OUR REPLY: : we kept our discussion but added a sentence at the end of section 2.4 that this is standard.
* * *
4- If e∈2Z for electric operators (Section 2.4), why is e∈Z allowed in Zc[g,1] (2.23)-(2.19)?

OUR REPLY: : Notation for Coulomb gas charge modified to avoid confusion and footnote 8 added.
* * *
5- It is strange that the partition function (2.23) is only partially described in Section 2.5, with one piece being postponed to eq. (2.31). Some restructuring would be welcome.

OUR REPLY: : We moved that last piece to section 2.5
* * *
6- 1 is not prime: is (2.32) valid for M=1?

OUR REPLY: : yes, sentence corrected
* * *
7- The notation Oe0+2P,0 is confusing, and obscures the fact that this operator is degenerate if P≥0. A notation of the type V1,P+1 would be much better. (I used the letter V in order to avoid the confusion with the non-diagonal operators Oe,m.) Similarly, in (2.31) the notation xe0+2P,0 is confusing.

OUR REPLY: : We introduced an alternative notation for operator Oe0+2P,0
* * *
8- When discussing the operator content, in other words when grouping the terms in the partition function into Virasoro characters, it would be good to systematically indicate which contributions come from which terms with which multiplicities. Presenting all the results as a table might be appropriate.

OUR REPLY: : we streamlined and clarified the discussion of how Virasoro characters form from various contributions. However we decided against adding a table.
* * *
9- Statements like 'does not exist as a primary operator' or 'a scalar operator of such dimension is present, but it's not a primary' or 'do not exist in the spectrum' should be made more precise.
In the presence of negative multiplicities, it is hard to say anything definite about the spectrum: having statements about characters would be more accurate. The appearance of degenerate characters suggests, but does not prove, that degenerate operators exist. And null vectors are both primary and descendent.

OUR REPLY: We believe our conclusions about the degenerate operators are correct. We added footnote 17 to clarify.
* * *
10- In Figure 7, at Q=2 there seem to be two degenerate operators of type (1,2) with different dimensions.

OUR REPLY: : One of these is on a solid line (critical), another dashed (tricritical). So they live in different theories.
* * *
11- The operators ϵ,ϵ′ are actually degenerate at all Q, whereas Figure 7 suggests that they are degenerate at Q=2,3 only. The indices (r,s) are not just positions in the Kac table: they make sense at all central charges.

OUR REPLY: : We are certainly aware that indices (r,s) make sense at all central charges, as our Eq. (2.31) says exactly that. We added a sentence to the caption of Figure 7 to make sure that this is not forgotten.
* * *
\subsubsection*{Section 4 before 4.1}

The authors could be more cautious in declaring that the light fields belong to degenerate conformal families: the characters suggest it, but do not prove it. Character-wise, a degenerate representation coincides with two non-degenerate representations with multiplicities 1 and −1, and we are in a context where negative multiplicities are allowed.

OUR REPLY: : this is partly addressed in footnote 17
* * *
Moreover, fixing three-point functions of degenerate fields at arbitrary central charges using crossing symmetry is not really complicated. The authors' extrapolation from minimal models is a circuitous route, which gives correct results because the correlators are uniquely determined by differential equations and crossing relations, as the authors correctly state. But this holds for all central charges.

OUR REPLY: . While we appreciate the referee’s remark, we do not feel obliged to act on it. The method we used, while it may look circuitous, was sufficient for our purposes and it gives consistent results which we conjecture to be correct. We hope that our results will stimulate further work upgrading these conjectures to the status of a theorem. We added two references to footnote 25 to indicate that alternative procedures exist
* * *
Therefore, I recommend the following:

12- Remove the discussion of minimal models, with its odd focus on the unitary models with integer m, whereas more general minimal models have two integer parameters. (The mention of minimal models in the article's Abstract could also be removed.)

OUR REPLY: while we followed many of the referee’s suggestions, we considered and decided against following this suggestion in light of our previous reply.
* * *
13- Give the explicit formulas for the structure constants, as this would allow readers to check their subsequent expansions around c=1. (See footnote 24.) The natural parameter for writing these structure constants is not m, but rather t or g. Maybe these formulas could be given in an appendix, which would include all needed structure constants, including those of the spin operator.

OUR REPLY: : While we followed many of the referee’s suggestions, we considered and decided against following this suggestion. In footnote 24 (current footnote 30) we give an explicit reference to the paper where the expression we used was given. We tend to believe there is no need to retype this expression.
* * *
\subsubsection*{Section 4.5}

This Section gives arguments for the existence of the real flow, based on the assumption that there exists a pair of complex conjugate CFTs that are moreover related by a RG flow. There are two arguments:

# A first-order argument argument, plus a heuristic argument that higher order corrections should deform the picture without changing it qualitatively.

# An all-orders argument, which is not rigorous because regularization issues are neglected, and which essentially shows the equivalence between the existence of the real flow, and the fixed point coupling constant gFP being pure imaginary.

The all-orders argument does not really establish the existence of the real flow, but it does constitute a nontrivial consistency check. Moreover, it illustrates the properties of correlation functions in complex CFTs.

The logic of the argument, and the technical details, are not clear enough.
Moreover, while interesting, this Section is not crucial to the main results of Section 4.
I recommend that Section 4.5 be either deleted, or clarified. Suggested clarifications include:

OUR REPLY: We kept the heuristic argument, but removed the all-order arguments which we agree was not clear enough.
* * *
Referee’s suggestions 14-19 are mute since we removed the all-order argument.
* * *
\subsubsection*{Miscellaneous suggestions}

20- Does eq. (2.4) for the partition function at arbitrary Q reduces to (2.2) when Q is integer?

OUR REPLY: yes, paragraph added at the end of Section 2.1 to stress that
* * *
21- In Subsection 2.2, clarify the statement that 'we will take an intuitive approach to symmetry for non-integer Q...'. Does this symmetry for non-integer Q play any role in the subsequent analysis?

OUR REPLY: a sentence added to subsection 2.2 to give two examples.
* * *
22- Typo: 'complex plain'.

OUR REPLY: done
* * *
23- In Section 3, the statement that 'multiplicities stay real for any Q' could be amended to 'multiplicities stay real for any real Q'. Thinking about complex Q is indeed quite natural in this context.

OUR REPLY: done. Related change: 3rd Paragraph of section 3 somewhat expanded.
* * *
24- There is a dual use of the notation g in Section 4.1: the original g from eq. (2.6), which appears again in eq. (4.7), differs from the g in eq. (4.9).

OUR REPLY: Indeed. We had footnote 22 about that , which we moved to the main text, right after 4.9, to decrease even further the chance for confusion.
* * *
25- In Section 4.1, the description of the Im-flip is confusing. Maybe an equation would be clearer.

OUR REPLY: the description of Im-flip is modified to make it hopefully less confusing
* * *
26- In the Conclusion, state explicitly what it means to 'make sense' for a Euclidean CFT, and for a Minkowskian CFT.

OUR REPLY: Adjusted, Hopefully more clear now.

OTHER CHANGES: We also streamlined discussion of drifting critical exponents Eqs 4.27-4.31 correcting a few typos. Conclusions are unchanged.

---

## Editorial Decision

published